



# Instrument Data Simulations for GRACE Follow-on: Observation and Noise Models

Neda Darbeheshti[1], Henry Wegener[1], Vitali Müller[1], Majid Naeimi[2], Gerhard Heinzel[1], and
Martin Hewitson[1]

[1]Max Planck Institute for Gravitational Physics (Albert Einstein Institute)-Leibniz Universität Hannover
[2]Institut für Erdmessung-Leibniz Universität Hannover

*Correspondence to:* Neda Darbeheshti (neda.darbeheshti@aei.mpg.de)

**Abstract.** The Gravity Recovery and Climate Experiment (GRACE) mission has yielded data on the Earth's gravity field to monitor temporal changes for more than fifteen years now. The GRACE twin satellites use microwave ranging with micrometer precision to measure distance variations between two satellites caused by the Earth's global gravitational field. GRACE Follow-on (GRACE-FO) will be the first satellite mission to use inter-satellite laser interferometry in space. The laser ranging instrument (LRI) will provide two additional measurements compared to the GRACE mission: interferometric inter-satellite ranging with nanometer precision and inter-satellite pointing information. We have designed a set of simulated GRACE-FO data, which include LRI measurements, apart from all other GRACE instrument data needed for the Earth's gravity field recovery. The simulated data files are publicly available via https://doi.org/10.22027/AMDC2 and can be used to derive gravity field solutions like from GRACE data. This paper describes the scientific basis and technical approaches used to simulate the GRACE-FO instrument data.

*Copyright statement.* TEXT

## 1 Introduction

The space gravimetry mission GRACE (Tapley et al., 2004) observes the Earth's gravity field changes with time. GRACE is the first low-low satellite-to-satellite tracking mission: the principal measurement is the distance variability between low orbit GRACE twin satellites which translates into the monthly gravity models (Wahr et al., 1998).

Kim (2000) published the first GRACE satellite simulation study before the launch of the GRACE satellites (in 2002). Now, seventeen years later, GRACE satellites are at the end of their lifetime and GRACE-FO data will be available soon. Although the GRACE-FO mission, and respectively its instrument data streams, will be very similar to GRACE, the necessity for GRACE-FO instrument data simulation emerges from the additional interferometric inter-satellite ranging. Most importantly, the operation of the LRI in addition to the primary K-band ranging (KBR) instrument yields extra information not only in the ranging measurement, but also in the attitude determination, since the LRI data stream will contain precise measurements of



the satellites' pitch and yaw angles. Exploitation of the new GRACE-FO measurements has great potential to improve spatial and temporal resolution of the Earth's gravity field solutions.

Also, there are different techniques to recover the Earth's gravity field from GRACE-like data (e.g., Reigber (1989), Gerlach et al. (2003), Mayer-Gürr (2006) , Rummel (1979)). Therefore, simulated instrument data provide a controlled, closed form
medium, to test and improve different gravity field recovery techniques.

We have generated a set of simulated GRACE-FO data for the period of one month. The data set is available for download via https://doi.org/10.22027/AMDC2. The main purpose of this paper is to describe the chain of instrument data simulation procedure. The first section presents the preliminaries for the data simulation, including the coordinates systems and symbols, followed by each section describing each instrument data simulation, including details of instruments' noise models.

## 2 Preliminaries

The following coordinate systems are used to define the various simulated data:

International Celestial Reference Frame (ICRF) – Inertial frame:

- – origin: center of mass (CoM) of the Earth
- – axes: according to IERS 2010 conventions (Petit and Luzum, 2010)

International Terrestrial Reference Frame (ITRF) – Earth-fixed (co-rotating) frame:

- – origin: CoM of the Earth
- – axes: according to IERS 2010 conventions (Petit and Luzum, 2010)

line-of-sight frame (LOSF), one per satellite, for GRACE A:

- – origin: satellite's CoM
- – $\boldsymbol{x}_{LOSF_A} = \frac{\boldsymbol{r}_B - \boldsymbol{r}_A}{|\boldsymbol{r}_B - \boldsymbol{r}_A|}$, where $\boldsymbol{r}$ is the satellites' position vector in the ICRF (i.e, roll axis).
- – $\boldsymbol{y}_{LOSF_A} = \frac{\boldsymbol{x}_{LOSF_A} \times \boldsymbol{r}_A}{|\boldsymbol{x}_{LOSF_A} \times \boldsymbol{r}_A|}$ (i.e, pitch axis)
- – $\boldsymbol{z}_{LOSF_A} = \boldsymbol{x}_{LOSF_A} \times \boldsymbol{y}_{LOSF_A}$ (i.e, yaw axis)
  (for GRACE B, A and B indices should be exchanged.)

satellite frame (SF), one per satellite according to Case et al. (2002):

- – origin: satellite's CoM





- $x_{SF}$ = from the origin to a target location of the phase center of the K/Ka band horn

- $y_{SF}$ = forms a right-handed triad with $x_{SF}$ and $z_{SF}$

- $z_{SF}$ = normal to $x_{SF}$ and to the plane of the main equipment platform, and positive towards the satellite radiator on the bottom of the GRACE-FO

The LOSF and SF are shown in Fig. 1. Since we did not model variations of the satellites' CoM (and the CoM coinciding with the on-board accelerometer's proof masses) for data simulation, the SF coincides with the science reference frame defined in Case et al. (2002).

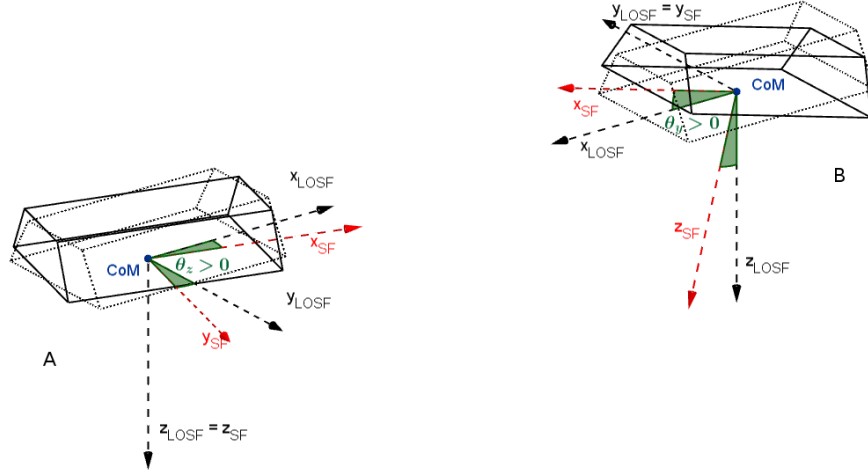

**Figure 1.** Illustration of SF and LOSF for GRACE satellites. Small positive yaw (left) and pitch (right) angles indicate the direction of rotation defining the sign of the pointing angles

All simulated data are published in GRACE Level-1B data format: daily files with 5-second sampling rate (cf. Case et al.,

10   2002). They can be considered pre-processed like GRACE Level-1B data. Time tags are given in GRACE GPS seconds, i.e. seconds since epoch 2000-01-01, 12:00:00 (no leap seconds applied). Five instrument data types were simulated:

- GPS Navigation Data (GNV1B)
  Simulated GPS positions and velocities are the output of the orbit integrator, which are rotated from ICRF to ITRF, and a GPS error is added to each. The error-free positions can be considered a kinematic orbit.

15   - K-Band Ranging System (KBR1B)
  Simulated KBR ranging data is derived from the error-free GPS positions and velocities with added KBR errors.




- – Star Camera (SCA1B)

  Simulated star camera quaternions are derived from the simulated roll, pitch and yaw angles with added errors.

- – Accelerometer (ACC1B)

  Simulated linear accelerations are calculated from the non-gravitational accelerations acting on the satellites. The error-free simulated star camera quaternions are used to transform the linear accelerations from ICRF to SF. The angular accelerations are calculated from the error-free simulated star camera quaternions.

- – Laser Ranging Instrument (LRI1B)

  Simulated LRI ranging data is derived from error-free GPS positions and velocities with added LRI errors.

Figure 2 shows a flowchart of the procedure used for the simulations. For each instrument, first the error-free observation was generated, and then the errors including instrument noise, bias and scale were applied to each instrument observation.

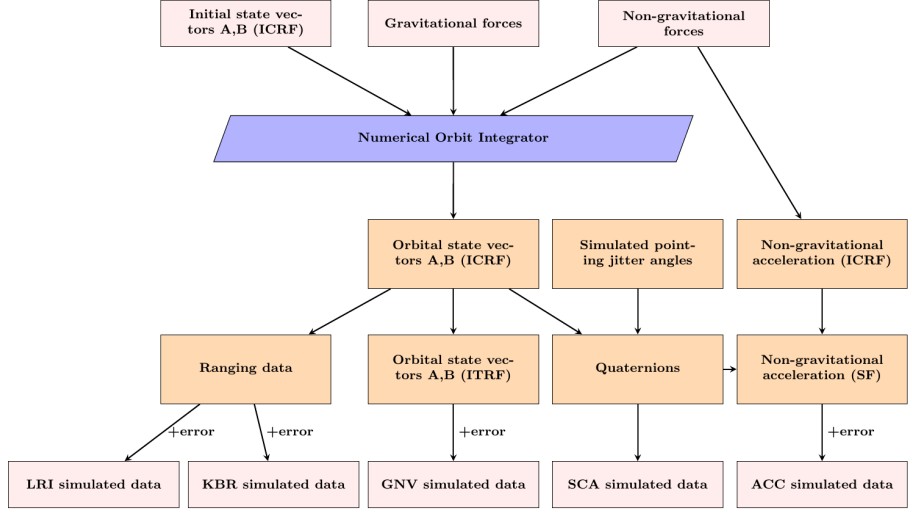

**Figure 2.** Flowchart of the simulation steps for GRACE-FO instrument data

In this paper,

- – The symbols $\delta$ and $\Delta$ are used for time-varying and constant errors, respectively.

- – The symbol $\tilde{\delta}$ denotes amplitude spectral densities (ASD).

For data simulations,

- – A five points numerical differentiation method was used for the numerical differentiations.





– The LISA Technology Package Data Analysis (LTPDA) toolbox (https://www.elisascience.org/ltpda/) for MATLAB
was used for generation of time series based on instrument noise models given in terms of ASD. LTPDA uses Franklin's
random noise generator method (Franklin, 1965) to generate arbitrarily long time series with a prescribed spectral density.

## 3 Simulating GNV1B Data

An orbit integrator is used to calculate the trajectories of the GRACE-FO satellites (GRACE-FO A and GRACE-FO B) by
numerical integration of Newton's second law of motion, based on knowledge of the forces acting on the satellite. Table 1
summarises the orbit integrator parameters.

**Table 1.** Orbit integrator parameters

| Parameter | Description |
| --- | --- |
| Numerical integration approach | Gauss-Jackson order 12 |
| Integration length | 31 days |
| Integration step size | 5 seconds |

The IERS2010 conventions are used for rotation between the ITRF and the international celestial reference frame ICRF.
Two types of force models were used for orbit integration:

Gravitational forces:

– A static gravity field of degree and order between 75 and 95.

– Ocean tide model (eot11a) up to degree and order 80.

– Direct tides of the Moon and Sun using NASA Jet Propulsion Laboratory (JPL) DE405 ephemeris.

– Anelastic solid Earth tides according to IERS2010.

Non-gravitational forces:

– Atmospheric drag model

– Solar radiation pressure model

The static gravity model and its degree and order are the unknowns for the gravity field recovery. The atmospheric drag and
solar radiation pressure models are described in Appendix A.



The input to the orbit integrator is the initial time and state (position and velocity vectors) of GRACE-FO A and GRACE-FO B at time 00:00:00, 2005-05-01. It calculates the two trajectories separately, beside the time series of accelerations along the trajectory from the gravitational and non gravitational force models. The output of the orbit integrator are the time series of the position, velocity and acceleration vectors of GRACE-FO A and GRACE-FO B:

$\boldsymbol{r}_A, \quad \dot{\boldsymbol{r}}_A, \quad \ddot{\boldsymbol{r}}_A, \quad \boldsymbol{r}_B, \quad \dot{\boldsymbol{r}}_B, \quad \ddot{\boldsymbol{r}}_B$

White noise with a level of a few $\frac{\mathrm{cm}}{\sqrt{\mathrm{Hz}}}$ was generated along $x$, $y$ and $z$ axes independently, and added to each satellite position:

$$\boldsymbol{r}_{GNV1B} = \boldsymbol{r} + \delta\boldsymbol{r}_{GNV1B} \tag{1}$$

Then the noise was differentiated numerically and added to the velocities along $x$, $y$ and $z$ axes separately for each satellite:

$$\dot{\boldsymbol{r}}_{GNV1B} = \dot{\boldsymbol{r}} + \delta\dot{\boldsymbol{r}}_{GNV1B} \tag{2}$$

## 4    Simulating SCA1B Data

The satellite attitude with respect to the ICRF is determined from the star cameras on board the satellites. The measured attitude is expressed in terms of quaternions $q$:

$$q = \begin{pmatrix} q_0 & q_1 & q_2 & q_3 \end{pmatrix} \tag{3}$$

Here, $q_0$ denotes the real component and $q_1$, $q_2$ and $q_3$ are the imaginary components of the quaternion. The time series of quaternions is provided in the SCA1B product.

In the GRACE satellites, an on-board attitude and orbital control system continuously attempts to align the pointing vector from the CoM to the K-band antenna phase center (APC) with the line-of-sight vector, in order to keep the geometric error in the KBR measurement as small as possible. However, the LRI measurements are subject to a very similar effect. More

importantly, even though the KBR will be the primary science instrument and the LRI, a technology demonstrator, a threshold of some milliradians pointing variations may not be exceeded to prevent the laser interferometer from falling out of lock. To the authors' knowledge, it is not decided yet how this problem shall be addressed in the best way. For this reason, we assumed that the satellites' attitude reference is the alignment of SF and LOSF.

Accordingly, satellite pointing angles can be computed from star camera quaternions and orbital positions (described in

Appendix B). For simulating star camera quaternions, one has to go the opposite way. Pointing angles from GRACE-FO attitude and orbital control system performance predictions were provided to us by JPL and AIRBUS Defense and Space. A model, which is based on the spectrum of these predicted angles, was used to simulate the pointing angles. The common approach for generating time series with a known spectrum is to use a random noise generator. Figure 3 shows the ASD of the simulated roll ($\theta_x$), pitch ($\theta_y$) and yaw ($\theta_z$) angles. One can see that all three angles have peaks mostly in the frequency

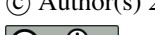



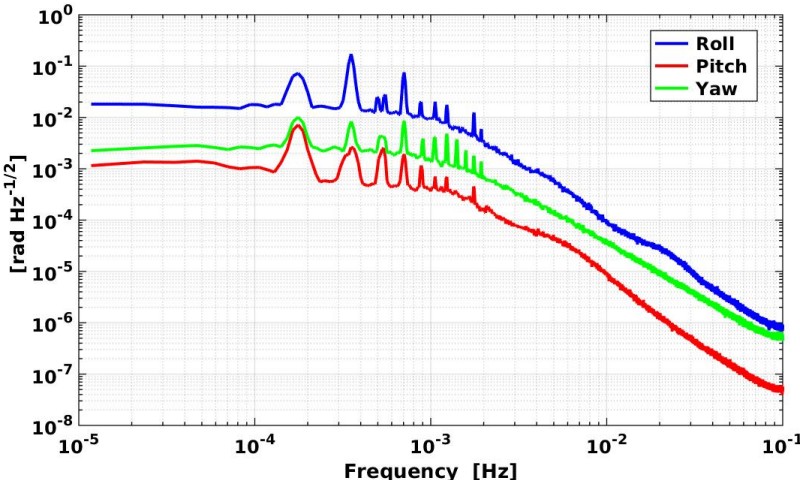

**Figure 3.** ASD of simulated roll, pitch and yaw angles

band between $10^{-4}$ and $2 \cdot 10^{-3}$. These peaks disturb the functionality of the random noise generator, thus they were modelled individually. The result is a time series of error-free inter-satellite pointing angles.

To simulate star camera measurements, white noise ($\delta\theta_{SCA1B}$) and a bias ($\Delta\theta_{SCA1B}$) was added to each error-free angle separately:

$$\theta_{x,SCA1B} = \theta_x + \delta\theta_{x,SCA1B} + \Delta\theta_{x,SCA1B}$$

$$\theta_{y,SCA1B} = \theta_y + \delta\theta_{y,SCA1B} + \Delta\theta_{y,SCA1B}$$

$$\theta_{z,SCA1B} = \theta_z + \delta\theta_{z,SCA1B} + \Delta\theta_{z,SCA1B} \tag{4}$$

Here, $\theta_x$, $\theta_y$ and $\theta_z$ are the error-free simulated roll, pitch and yaw angles; $\theta_{x,SCA}$, $\theta_{y,SCA}$ and $\theta_{z,SCA}$ are simulated star camera roll, pitch and yaw angles.

The GRACE-FO satellites are equipped with improved star cameras compared to GRACE, and the number of star camera heads will increase from two to three per satellite (Gath, 2016); also Bandikova et al. (2012) suggested that proper combination of the different star camera heads reduces high frequency noise of the pointing angles. Accordingly, it is expected that a better estimation of pointing angles from GRACE-FO star camera data will be available. Therefore, white noise with a level of a few ten $\frac{\mu\text{rad}}{\sqrt{\text{Hz}}}$ was chosen, which is lower than the current noise level in roll, pitch and yaw angels estimated from the GRACE star camera data. The value of bias for each angle was chosen in range of a few milliradians (Horwath et al., 2011). Figure 4 shows simulated star camera roll, pitch and yaw angles.



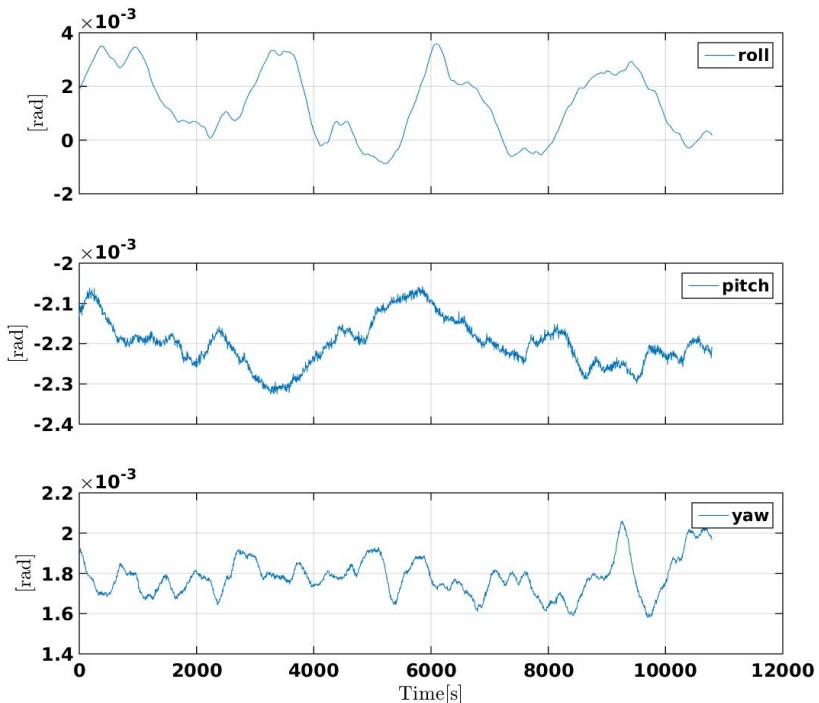

**Figure 4.** Simulated star camera roll, pitch and yaw angles during two orbital revolutions for GRACE-FO A

From the contaminated simulated pointing angles of equations (4), the rotation matrix $\mathbf{R}_{SF}^{LOSF}$ was built; and with the error-free simulated orbit positions, the rotation matrix $\mathbf{R}_{LOSF}^{ICRF}$ was built; having these two matrices, the matrix $\mathbf{R}_{SF}^{ICRF}$ is

$$\mathbf{R}_{SF}^{ICRF} = \mathbf{R}_{LOSF}^{ICRF} \cdot \mathbf{R}_{SF}^{LOSF}, \tag{5}$$

containing the simulated star camera quaternions (cf. Fig. 5). Finally, the simulated quaternions can be recovered from the
5   rotation matrix $\mathbf{R}_{SF}^{ICRF}$ by using the equations (Wu et al., 2006):

$$q_0 = \frac{1}{2}\sqrt{1 + R_{11} + R_{22} + R_{33}}$$
$$q_1 = \frac{(R_{23} - R_{32})}{4q_0}$$
$$q_2 = \frac{(R_{31} - R_{13})}{4q_0}$$
$$q_3 = \frac{(R_{12} - R_{21})}{4q_0} \tag{6}$$





where $R_{ij}$ are the elements of $\mathbf{R}_{SF}^{ICRF}$. Note that the equations 6 are only numerically stable, as long as the trace of $\mathbf{R}$ is non-negative (i.e. not close to $-1$). A numerically stable pseudocode that was used is shown in Appendix C.

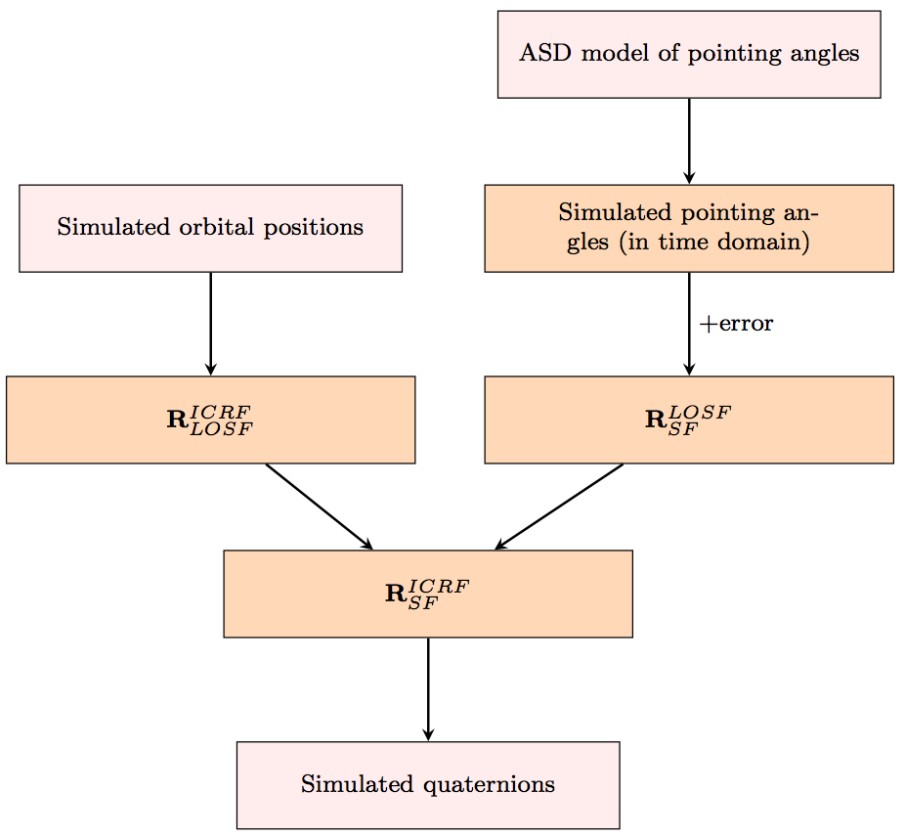

**Figure 5.** Flowchart of the simulation steps for SCA1B data

Two other sets of quaternions were generated. Error-free quaternions from error-free pointing angles in equations (4); and noisy quaternions that come from white noise contaminated pointing angles without the bias. We will refer to these two set of quaternions in the following sections.

## 5  Simulating ACC1B Data

Figure 2 shows that the non-gravitational accelerations were computed along the orbit in ICRF.





## 5.1 Linear Accelerations

The non-gravitational accelerations are the sum of atmospheric drag and solar radiation pressure accelerations (cf. Appendix A) along the orbit in inertial frame (ICRF). The non-gravitational accelerations $\ddot{\boldsymbol{r}}^{ICRF}$ were transformed into the satellite frame $\ddot{\boldsymbol{r}}^{SF}$ using the rotation matrix $\mathbf{R}^{SF}_{ICRF}$ from error-free simulated quaternions:

$$\ddot{\boldsymbol{r}}^{SF} = \mathbf{R}^{SF}_{ICRF} \cdot \ddot{\boldsymbol{r}}^{ICRF} \tag{7}$$

After being transformed into the SF, the linear accelerations were multiplied by the scale factors $s_x$, $s_x$ and $s_z$, and then the accelerometer noise time series ($\delta\ddot{\boldsymbol{r}}_{ACC1B}$) and the biases ($\Delta\ddot{\boldsymbol{r}}_{ACC1B}$) were added along $x$, $y$ and $z$ axes independently:

$$\ddot{\boldsymbol{r}}^{SF}_{ACC1B} = \begin{bmatrix} s_x & 0 & 0 \\ 0 & s_y & 0 \\ 0 & 0 & s_z \end{bmatrix} \cdot \ddot{\boldsymbol{r}}^{SF} + \delta\ddot{\boldsymbol{r}}_{ACC1B} + \Delta\ddot{\boldsymbol{r}}_{ACC1B} \tag{8}$$

The ASD noise model of Kim (2000) was used to generate accelerometer noise ($\delta\ddot{\boldsymbol{r}}_{ACC1B}$):

$$\begin{aligned} \tilde{\delta\ddot{\boldsymbol{r}}}_{x/z,ACC1B}(f) &= 10^{-10} \cdot \sqrt{1 + \frac{0.005\mathrm{Hz}}{f}} \frac{\mathrm{m/s}^2}{\sqrt{\mathrm{Hz}}} \qquad 10^{-5} \leq f \leq 10^{-1} \\ \tilde{\delta\ddot{\boldsymbol{r}}}_{y,ACC1B}(f) &= 10^{-9} \cdot \sqrt{1 + \frac{0.1\mathrm{Hz}}{f}} \frac{\mathrm{m/s}^2}{\sqrt{\mathrm{Hz}}} \qquad 10^{-5} \leq f \leq 10^{-1} \end{aligned} \tag{9}$$

The $y$ axis in SF ($\boldsymbol{y}_{SF}$ in Fig. 1) is considered the least sensitive axis for accelerometer measurements (Kim, 2000). The noise ASD of the sensitive axes and the less sensitive axis are shown in Fig. 6. One month time series of the accelerometer noise was generated separately for $x$, $y$ and $z$ axes from the ASD models and added to the accelerations (equation (8)). Values close to GRACE accelerometer scale and bias along each axis were chosen, and kept constant for the one month of the simulated data. Therefore, in total for both satellites, six accelerometer scale parameters and six accelerometer bias parameters should be estimated during the gravity field recovery using one month of the simulated data.

## 5.2 Angular Accelerations

The error-free simulated quaternions were used to generate angular accelerations, based on the relations between the quaternions and angular accelerations ($\dot{\omega}_x, \dot{\omega}_y, \dot{\omega}_z$) (cf. Müller, 2010):

$$\begin{bmatrix} \dot{\omega}_x \\ \dot{\omega}_y \\ \dot{\omega}_z \\ -2\Sigma\dot{q}_m^2 \end{bmatrix} = 2 \cdot \begin{bmatrix} -q_1 & q_0 & q_3 & -q_2 \\ -q_2 & -q_3 & q_0 & q_1 \\ -q_3 & q_2 & -q_1 & q_0 \\ q_0 & q_1 & q_2 & q_3 \end{bmatrix} \cdot \begin{bmatrix} \ddot{q}_0 \\ \ddot{q}_1 \\ \ddot{q}_2 \\ \ddot{q}_3 \end{bmatrix}$$

where $\ddot{q}_m$ are the numerically differentiated simulated quaternions.

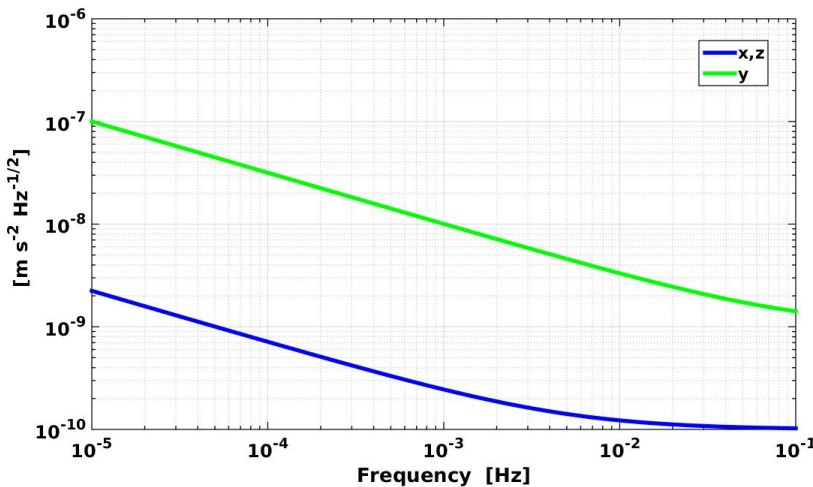

**Figure 6.** ASD of accelerometer noise

## 6  Simulating KBR1B Data

The position, velocity and acceleration differences of GRACE-FO A and GRACE-FO B are calculated as follows:

$$\boldsymbol{r}_{AB} = \boldsymbol{r}_B - \boldsymbol{r}_A$$
$$\dot{\boldsymbol{r}}_{AB} = \dot{\boldsymbol{r}}_B - \dot{\boldsymbol{r}}_A$$
$$\ddot{\boldsymbol{r}}_{AB} = \ddot{\boldsymbol{r}}_B - \ddot{\boldsymbol{r}}_A, \tag{10}$$

in order to calculate simulated error-free range, range rate and range acceleration according to:

$$\rho = \sqrt{\boldsymbol{r}_{AB} \cdot \boldsymbol{r}_{AB}} \tag{11}$$

$$\dot{\rho} = \frac{\boldsymbol{r}_{AB}}{\rho} \cdot \dot{\boldsymbol{r}}_{AB} \tag{12}$$

$$\ddot{\rho} = -\frac{\dot{\rho}^2}{\rho} + \frac{\dot{\boldsymbol{r}}_{AB} \cdot \dot{\boldsymbol{r}}_{AB}}{\rho} + \frac{\boldsymbol{r}_{AB}}{\rho} \cdot \ddot{\boldsymbol{r}}_{AB} \tag{13}$$

where $\cdot$ is the vector dot product.

The GRACE-FO KBR instrument (as in GRACE) will measure the biased range between the twin satellites; respectively, a bias ($\Delta\rho$) of a few centimeters was added to the error-free range ($\rho$). The KBR instrument noise is dominated by oscillator and system noise ($\delta\rho_{SO}$). It was added to the error-free ranging products, as well as a geometric error, which is a pointing jitter



coupling effect caused by an offset of the KBR antenna phase center for each satellite A and B ($\delta\rho_{APC}$):

$$\rho_{KBR1B} = \rho + \delta\rho_{SO} + \delta\rho_{APC_A} + \delta\rho_{APC_B} + \Delta\rho_{KBR1B}$$

$$\dot{\rho}_{KBR1B} = \dot{\rho} + \delta\dot{\rho}_{SO} + \delta\dot{\rho}_{APC_A} + \delta\dot{\rho}_{APC_B}$$

$$\ddot{\rho}_{KBR1B} = \ddot{\rho} + \delta\ddot{\rho}_{SO} + \delta\ddot{\rho}_{APC_A} + \delta\ddot{\rho}_{APC_B} \qquad (14)$$

In the following, these two error sources are described.

### 6.1 System and Oscillator Noise

5 The following ASD model was used to generate KBR noise:

$$\tilde{\delta\rho}_{SO}(f) = 10^{-6} \cdot \sqrt{1 + \left(\frac{0.0018\,\mathrm{Hz}}{f}\right)^4}\,\frac{\mathrm{m}}{\sqrt{\mathrm{Hz}}} \qquad 10^{-5} \leq f \leq 10^{-1} \qquad (15)$$

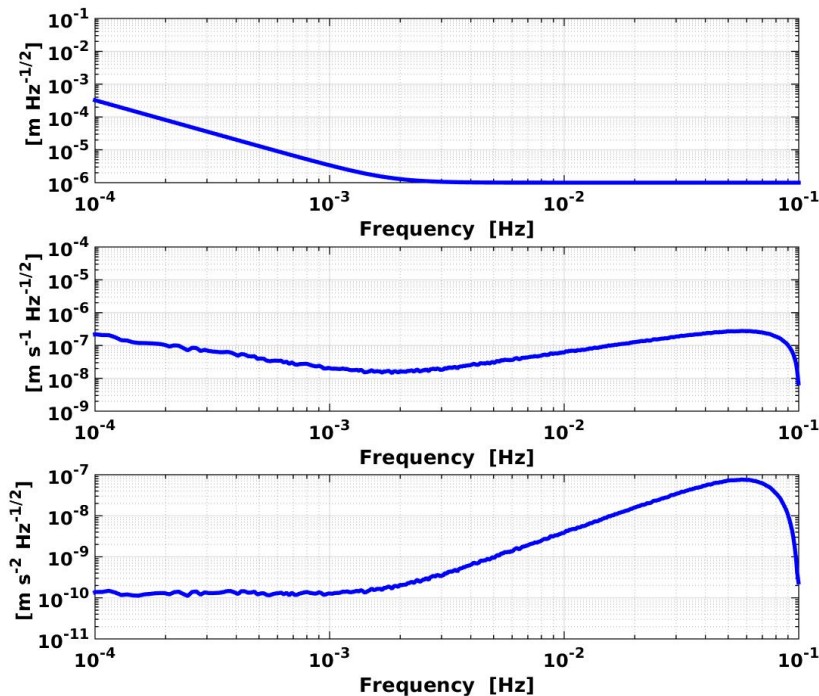

**Figure 7.** ASD of KBR oscillator and system noise for range (top), range rate (middle) and range acceleration (bottom)

This ASD model is in agreement with the system and oscillator KBR noise for the satellite pair separation of 238 km in Kim (2000). Figure 7 illustrates the ASD model. Based on this model, one month time series of the range noise was generated. Then





numerical differentiation was used to generate range rate noise and range acceleration noise from the range noise time series (cf. Fig. 8).

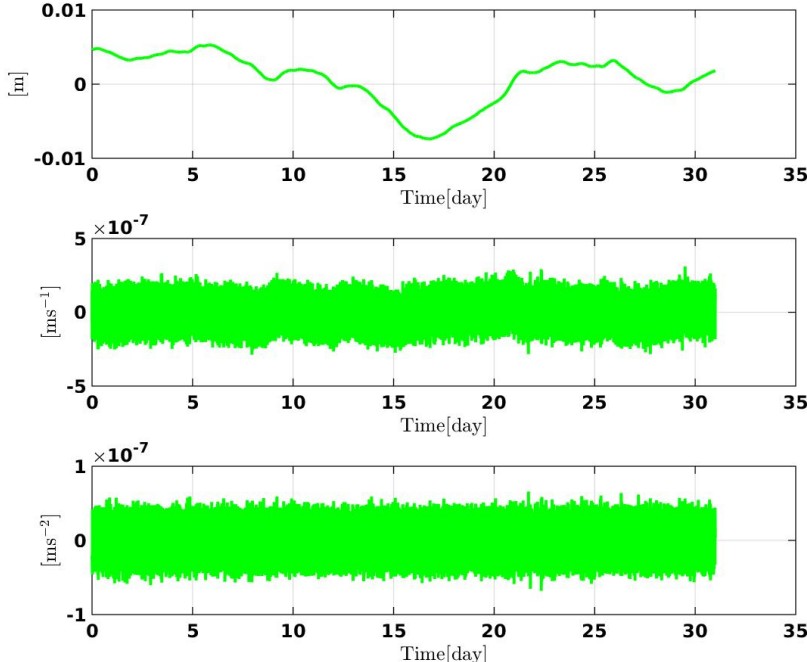

**Figure 8.** Time series of KBR oscillator and system noise for range (top), range rate (middle) and range acceleration (bottom)

## 6.2 Antenna Phase Center Pointing Jitter Coupling

The KBR instrument measures the distance between the antenna phase centers, which are placed nominally on the SF x-axis,

5    almost $1.5$ m away from the satellites' CoM. However, due to manufacturing imperfections and due to the large acceleration of the system during launch, the actual positions differ from the nominal ones. Consequently, any pointing jitter (deviations of the satellites' attitudes from their nominal attitudes) causes a geometric error in the ranging measurement. In the absence of such misplacements and in the absence of pointing jitter, this effect would be zero (rather, constant) and hence not effect the measured (biased) range. Given the antenna phase center (APC) offset vector $(p_A^{SF})$ in SF and the matrix rotating from SF to

10    ICRF, this error is computed as:

$$\delta\rho_{APC_A} = -\left(e_{AB}^{ICRF}\right)^T \cdot \mathbf{R}_{SF}^{ICRF} \cdot p_A^{SF}, \tag{16}$$





i.e. it is the APC offset (w.r.t. CoM) projected onto the line-of-sight. For the simulation, the $\mathbf{R}_{SF}^{ICRF}$ was calculated from equation (B1) using the error-free simulated quaternions, and the line-of-sight vector ($e_{AB}^{ICRF}$) was calculated from the error-free satellite positions in ICRF:

$$e_{AB}^{ICRF} = \frac{\boldsymbol{r}_B - \boldsymbol{r}_A}{|\boldsymbol{r}_B - \boldsymbol{r}_A|} \qquad (17)$$

For GRACE-FO B, the indices A and B should be swapped in equations (16) and (17). Figure 9 shows time series of the APC offset pointing jitter coupling for one month of GRACE-FO A.

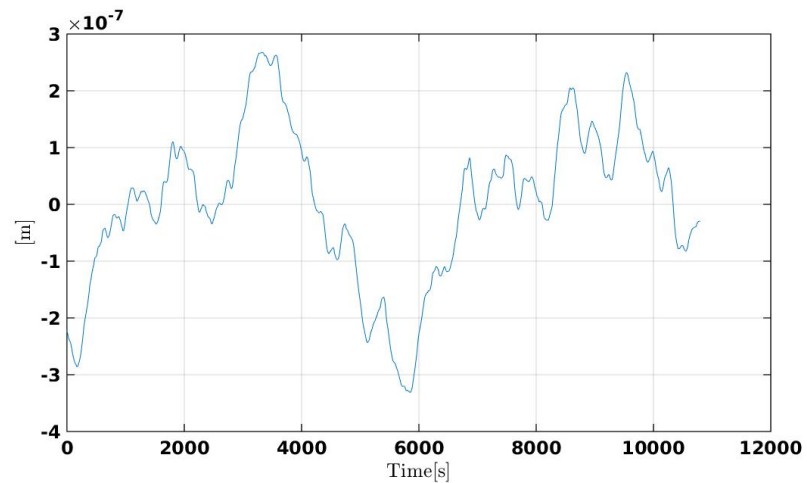

**Figure 9.** Time series of APC offset pointing jitter coupling with subtracted mean value, during two orbital revolutions for GRACE-FO A

In GRACE, there have been calibration manoeuvers in order to try and estimate the APC offset vectors ($\boldsymbol{p}_A^{SF}, \boldsymbol{p}_B^{SF}$). The estimates have been published by JPL in the VKB1B files (Case et al., 2002). For the simulations, values of similar magnitude

were chosen. These values are not directly given to the user, however, the simulated KBR1B files include a column of simulated estimated correction terms. This means that it is computed from the imperfect attitude information that is provided via simulated SCA1B files. Real GRACE KBR1B data also contains this column, which is called antenna offset correction (AOC) term (cf. Case et al., 2002) . It has to be added to the KBR ranging measurement, so it describes the negative of the error term:

$$AOC_\rho \approx -\delta\rho_{APC_A} - \delta\rho_{APC_B}. \qquad (18)$$

For the simulations, the correction term of $AOC_\rho$ was computed according to equation (16), with the difference that the matrix $\mathbf{R}_{SF}^{ICRF}$ was derived from the simulated noisy quaternions without the bias.





A second and third column is also provided, computed by numerical differentiation, describing the correction for range rate and range acceleration:

$$AOC_{\dot{\rho}} \approx -\delta\dot{\rho}_{APC_A} - \delta\dot{\rho}_{APC_B}$$
$$AOC_{\ddot{\rho}} \approx -\delta\ddot{\rho}_{APC_A} - \delta\ddot{\rho}_{APC_B}. \tag{19}$$

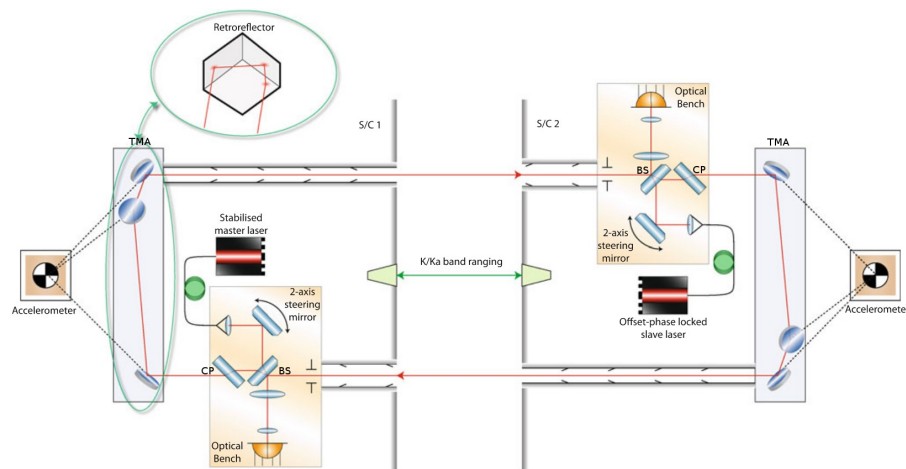

**Figure 10.** GRACE-FO laser ranging instrument optical layout (from Sheard et al. (2012)). BS beamsplitter, CP compensation plate, TMA triple mirror assembly

## 7  Simulating LRI1B Data

The structure of the LRI1B data file is similar to the KBR1B file, but it contains two additional observations of pitch and yaw angles. Tables D1 and D2 in Appendix D show the format of the data records for simulated KBR1B and LRI1B files. The simulated error-free range, range rate and range accelerations are calculated from the equations (11), (12) and (13). Apart from a bias of a few centimeters, various other errors were added:

$$\rho_{LRI1B} = \alpha \cdot (\rho + \delta\rho_{LF} + \delta\rho_{TMA_A} + \delta\rho_{TMA_B} + \delta\rho_{ALQ_A} + \delta\rho_{ALQ_B} + \Delta\rho_{LRI1B})$$
$$\dot{\rho}_{LRI1B} = \alpha \cdot (\dot{\rho} + \delta\dot{\rho}_{LF} + \delta\dot{\rho}_{TMA_A} + \delta\dot{\rho}_{TMA_B} + \delta\dot{\rho}_{ALQ_A} + \delta\dot{\rho}_{ALQ_B})$$
$$\ddot{\rho}_{LRI1B} = \alpha \cdot (\ddot{\rho} + \delta\ddot{\rho}_{LF} + \delta\ddot{\rho}_{TMA_A} + \delta\ddot{\rho}_{TMA_B} + \delta\ddot{\rho}_{ALQ_A} + \delta\ddot{\rho}_{ALQ_B}) \tag{20}$$

In equations (20), $\alpha = 1 + 10^{-6}$ is a scale factor which is due to the limited accuracy of the absolute laser frequency value for the phase to length conversion. Three main LRI noise sources in equations (20) are: laser frequency (LF) noise ($\delta\rho_{LF}$), and the coupling of the pointing jitter into the length measurement via triple mirror assembly (TMA) (Fig. 10) for each satellite A and B ($\delta\rho_{TMA}$) and the additional linear and quadratic pointing jitter coupling ($\delta\rho_{ALQ}$). This is a selection of relatively



well-known LRI error sources, where LF and TMA errors are expected to be the dominating ones. For the range rate and range acceleration noise, the errors were numerically differentiated. In the following, the LRI error sources are described in detail.

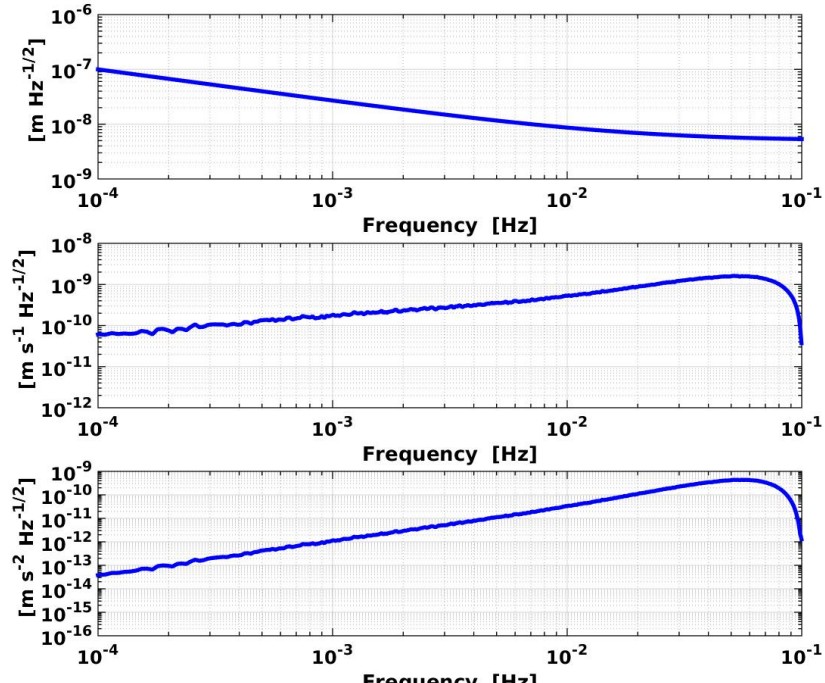

**Figure 11.** ASD of laser frequency noise for range (top), range rate (middle) and range acceleration (bottom)

## 7.1 Laser Frequency Noise

Based on LRI cavity performance tests carried out by JPL, the current best estimate of the ASD of the laser frequency noise 5 (i.e., the ranging noise which is induced by frequency jitter of the LRI master laser) for a satellite separation of 238 km is:

$$\tilde{\delta}\rho_{LF}(f) = 5 \cdot 10^{-9} \cdot \sqrt{1 + \left(\frac{0.0182\text{Hz}}{f}\right)^2} \; \frac{\text{m}}{\sqrt{\text{Hz}}} \tag{21}$$

Figure 11 illustrates this noise. Note that this specific ASD corresponds to a constant satellite separation (of 238 km), which is a sufficient simplification for the purpose of generating noise time series.

10     One month time series of the range noise $\delta\rho_{LF}$ was generated from the ASD model (cf. Fig. 11). Then numerical differentiation was used to generate range rate noise and range acceleration noise from the noise range time series (cf. Fig. 12).



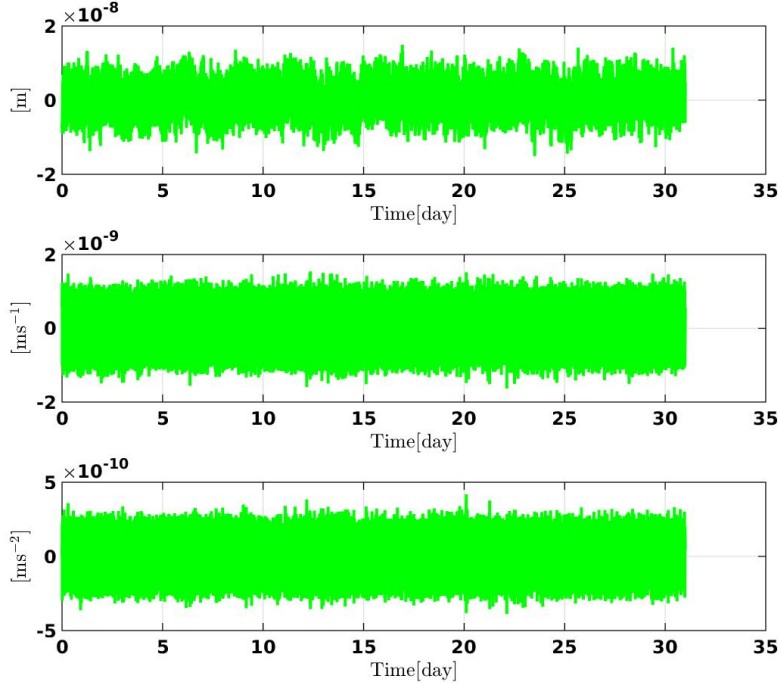

**Figure 12.** Time series of laser frequency noise for range (top), range rate (middle) and range acceleration (bottom)

## 7.2 Triple Mirror Assembly Pointing Jitter Coupling

With a good approximation, the LRI measures the biased distance between the TMA vertices of the twin satellites (Fig. 10). Both the pointing jitter and frame misalignments couple into the LRI ranging measurement. This effect is in principal the same as the geometric error effect due to the APC position in the KBR measurement. The only difference is that the nominal positions of the TMA vertices lie in the CoM, whereas the nominal positions of the APC are almost $1.5$ m in SF-x-direction away from this point.

An offset of the TMA vertex from the satellites'CoM leads to the coupling of satellite pointing jitter into the round trip length variations measured by the LRI (Fig. 13). The magnitudes of TMA vertex offset vectors ($\boldsymbol{v}^{SF}$) along $x$, $y$ and $z$ axes were chosen in the order of a few hundred micrometers. The real values after the GRACE-FO launch are unknown and will have to be calibrated. To calculate $\delta\rho_{TMA}$, the TMA vertex offset vector ($\boldsymbol{v}^{SF}$) is rotated from the SF into the ICRF and then projected onto the line-of-sight:

$$\delta\rho_{TMA_A} = -\left(e_{AB}^{ICRF}\right)^T \cdot \mathbf{R}_{SF}^{ICRF} \cdot \boldsymbol{v}_A^{SF} \tag{22}$$




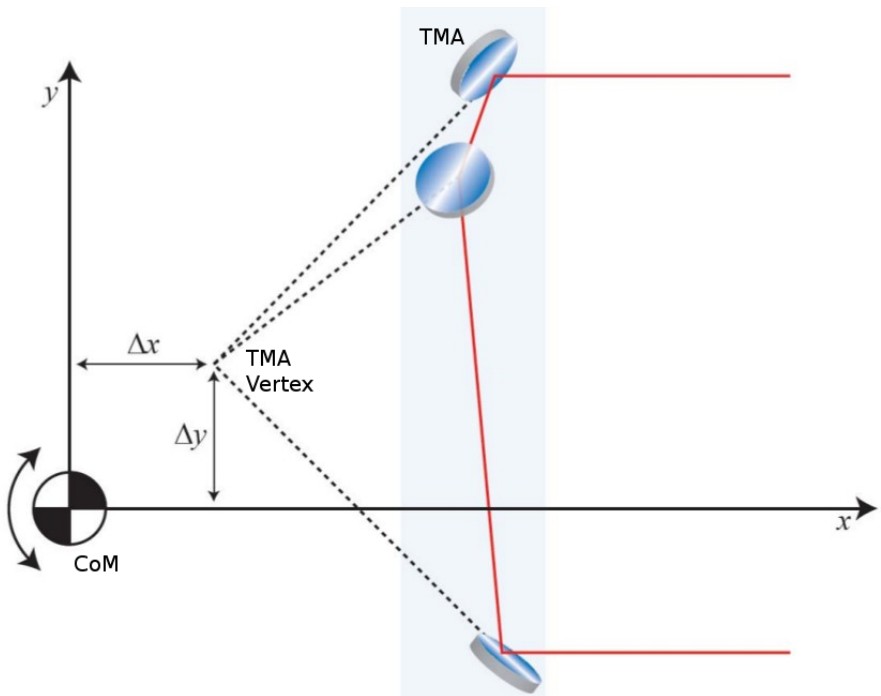

**Figure 13.** Triple mirror assembly vertex offset from the satellites' center of mass in two dimensions

where $e_{AB}^{ICRF}$ is the line-of-sight vector in ICRF. Again, for GRACE-FO B, indices A and B should be swapped. Figure 14 shows time series of TMA pointing jitter coupling for one month of GRACE-FO A.

Similar to the KBR1B files, LRI1B files contain correction terms - vertex point correction (VPC) terms - for range, range rate and range acceleration, which were calculated using the simulated noisy quaternions without the bias:

$$VPC_\rho \approx -\delta\rho_{TMA_A} - \delta\rho_{TMA_B}$$
$$VPC_{\dot\rho} \approx -\delta\dot\rho_{TMA_A} - \delta\dot\rho_{TMA_B}$$
$$VPC_{\ddot\rho} \approx -\delta\ddot\rho_{TMA_A} - \delta\ddot\rho_{TMA_B}.$$

(23)

### 7.3 Additional Linear and Quadratic Pointing Jitter Coupling

There is additional linear and quadratic coupling (ALQ) of the pointing jitter angles ($\theta_x$, $\theta_y$ and $\theta_z$) into the length measurements, which can be described as:

$$\delta\rho_{ALQ_A} = \begin{bmatrix} c_{x_A} & c_{y_A} & c_{z_A} \end{bmatrix} \cdot \begin{bmatrix} \theta_{x_A} \\ \theta_{y_A} \\ \theta_{z_A} \end{bmatrix} + \begin{bmatrix} \theta_{x_A} & \theta_{y_A} & \theta_{z_A} \end{bmatrix} \cdot \begin{bmatrix} c_{xx_A} & c_{xy_A} & c_{xz_A} \\ 0 & c_{yy_A} & c_{yz_A} \\ 0 & 0 & c_{zz_A} \end{bmatrix} \cdot \begin{bmatrix} \theta_{x_A} \\ \theta_{y_A} \\ \theta_{z_A} \end{bmatrix}$$

(24)



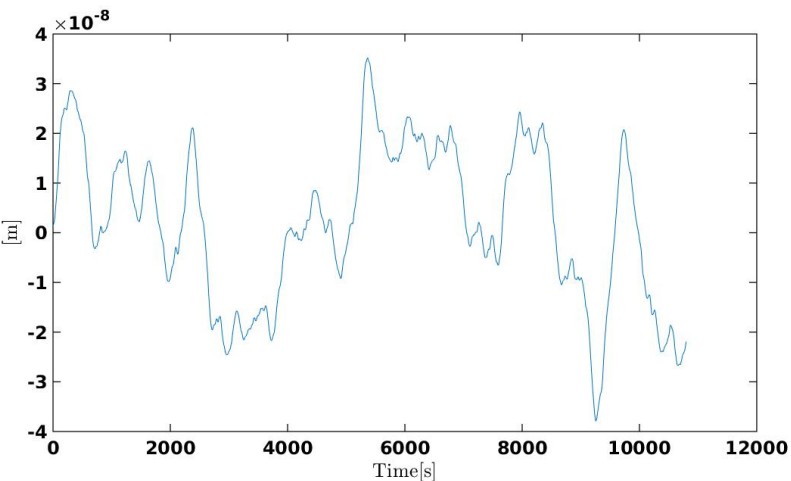

**Figure 14.** Time series of TMA pointing jitter coupling with subtracted mean value, during two orbital revolutions for GRACE-FO A

For GRACE-FO B, the indices A should be exchanged into B in equation (24). Linear coefficients of $c_x$, $c_y$ and $c_z$ are estimated to be in the order of a few $\frac{\mu m}{rad}$ and quadratic coefficients of $c_{xy}$, $c_{xz}$ and . . . are in the order of a few $\frac{mm}{rad^2}$. Error-free time series of $\theta_x, \theta_y, \theta_z$ were used to simulate $\delta\rho_{ALQ}$. Figure 15 shows time series of ALQ pointing jitter coupling for one month of GRACE-FO A.

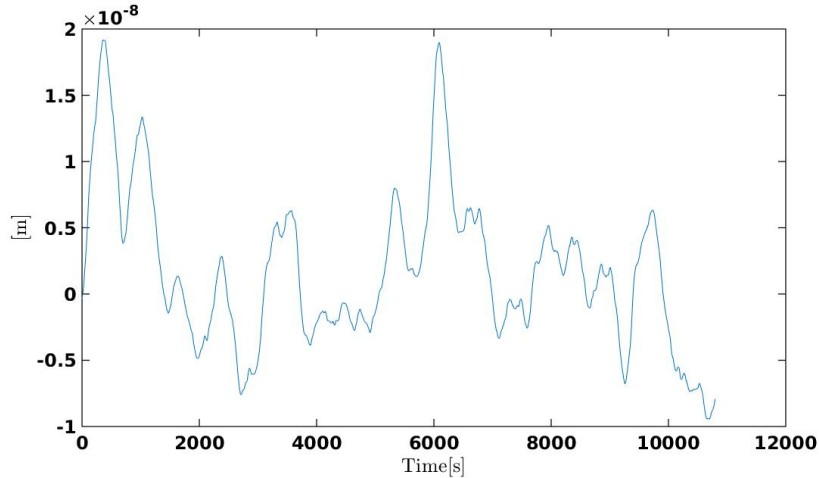

**Figure 15.** Time series of ALQ pointing jitter coupling with subtracted mean value, during two orbital revolutions for GRACE-FO A

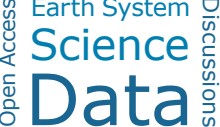

### 7.4 Differential Wavefront Sensing: Pitch And Yaw Measurements

Differential wavefront sensing (DWS) is a well known technique for measuring the relative wavefront misalignment between two laser beams with high sensitivity (Sheard et al., 2012). Figure 16 illustrates the basic principal of DWS. DWS provides two extra measurements of the satellite attitude: yaw and pitch pointing angles with respect to the line-of-sight.

DWS angle measurements on board GRACE-FO are obtained from the steering mirror on the LRI optical bench (Sheard et al., 2012). The steering mirror orientation is controlled using the DWS error signals, constantly driving the error signals back to zero. The steering mirror orientation is recorded as pitch and yaw angles. However, the steering mirror can only turn in quanta of $4.5\mu$rad around the pitch axis and $6\mu$rad around the yaw axis. Therefore, the angle determination is quantization

10   limited.

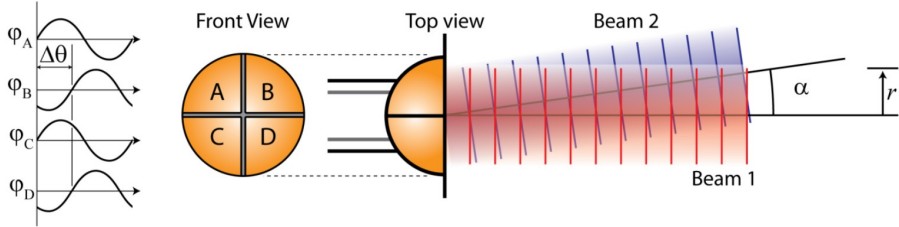

**Figure 16.** Differential wavefront sensing principal. Two beams of radius $r$ with a relative wavefront tilt of $\alpha$ are detected by a quadrant photodetector. The two beams also have a slight frequency difference (from Sheard et al. (2012))

For each satellite, DWS pitch and yaw measurements were simulated by:

$$\theta_{y,DWS} = \mathrm{round}\left(\frac{\theta_y}{4.5 \cdot 10^{-6}\mathrm{rad}}\right) \cdot 4.5 \cdot 10^{-6}\mathrm{rad} + \Delta\theta_{y,DWS}$$

$$\theta_{z,DWS} = \mathrm{round}\left(\frac{\theta_z}{6.0 \cdot 10^{-6}\mathrm{rad}}\right) \cdot 6.0 \cdot 10^{-6}\mathrm{rad} + \Delta\theta_{y,DWS}, \tag{25}$$

where "round" means rounding towards nearest integer. $\theta_{y,DWS}$ and $\theta_{z,DWS}$ are the simulated DWS pitch and yaw angles, and $\theta_y$ and $\theta_z$ are the error-free pitch and yaw angles. The biases ($\Delta\theta_{y,DWS}, \Delta\theta_{y,DWS}$) stem mainly from a misalignment of

15   the LRI frame with respect to the SF, which is expected to be within the range of few milliradians. Figure 17 shows simulated DWS pitch and yaw angles.



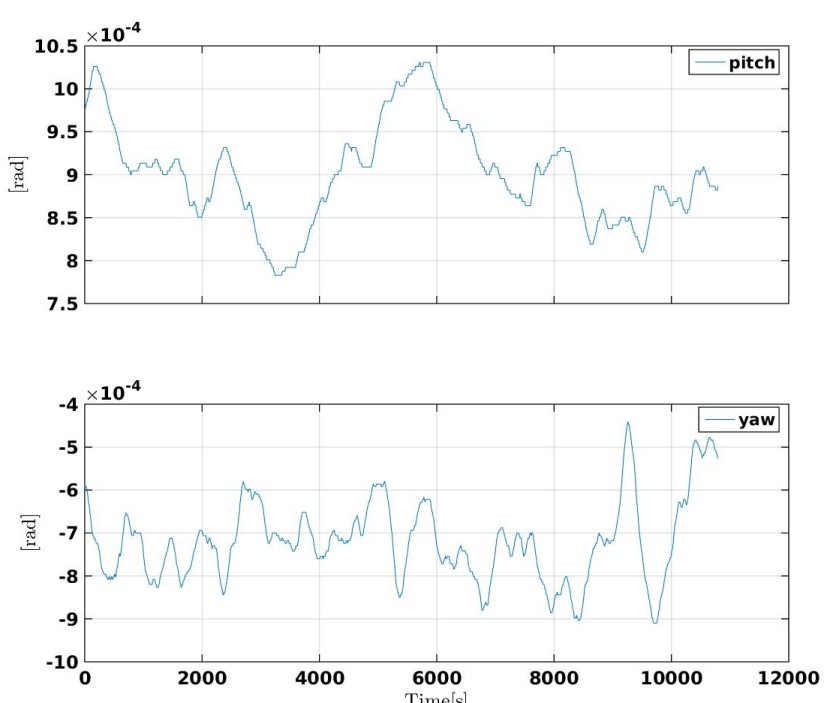

**Figure 17.** Simulated DWS pitch and yaw angles during two orbital revolutions for GRACE-FO A



## 8 Conclusions

The quality of the temporal gravity field solutions from the GRACE-FO mission depends on its multi-sensor system consisting of inter-satellite ranging with microwave and laser ranging instrument, GPS orbit tracking, accelerometry, and attitude sensing. In this paper, the simulation of observations and the noise models for GRACE-FO major instruments were described. For the

first time, simulated LRI data that includes DWS attitude information were generated. The simulated LRI ranging and attitude data may be used in different data analysis scenarios for GRACE-FO, such as combination of KBR and LRI data, calibration, or estimation of geometric corrections for both KBR and LRI ranging.

On the other hand, different Earth's gravity field solutions derived from actual satellite data can only be compared against each other, because the real Earth's gravity field is not known. This is a major problem in the evaluation of the performance

of gravity field recovery approaches. A closed-loop simulation starting with a known gravity field provides the opportunity to overcome this problem by comparing the input gravity field and the gravity field solutions. Also, the effect of instrument noise on gravity field solutions can be investigated by comparing observation residuals with the simulated instrument noise.

*Data availability.* The simulated instrument GRACE-FO data are available via https://doi.org/10.22027/AMDC2.

## Appendix A

In the following, we briefly describe the models for atmospheric drag and solar radiation pressure, that were used for the orbit simulations.

### A1  Atmospheric Drag Model

The acceleration due to atmospheric drag is calculated with the following formula from aerodynamic theory (e.g., Montenbruck and Gill, 2000, p. 84), with $A$ the satellite's cross sectional area, $m$ the mass of the satellite, $e_v$ the unit vector of the velocity

relative to the atmosphere, $C_D$ the drag coefficient and $p$ the atmospheric density at the location of the satellite:

$$\ddot{\boldsymbol{r}} = -\frac{1}{2} C_D \frac{A}{m} p v_r^2 \cdot \boldsymbol{e}_v \tag{A1}$$

For the calculation of the relative velocity $\boldsymbol{v}_r$, the assumption is made that the atmosphere co-rotates with the Earth. This leads to:

$$\boldsymbol{v}_r = \dot{\boldsymbol{r}} - \boldsymbol{\omega}_\oplus \times \boldsymbol{r},$$

with $\dot{\boldsymbol{r}}$ the inertial velocity vector of the satellite, $\boldsymbol{r}$ the position vector and $\boldsymbol{\omega}_\oplus$ the Earth's angular velocity. $v_r^2$ in formula A1 is then the square of the absolute relative velocity,

$$v_r^2 = |\boldsymbol{v}_r|^2.$$



## A2 Solar Radiation Pressure Model

A satellite exposed to radiation from the Sun experiences a force arising from the absorption and reflection of incident photons. The resulting acceleration was modelled by:

$$\ddot{\boldsymbol{r}} = -\nu \frac{P_\odot}{m} \left( \frac{AU}{x} \right)^2 \cdot \sum_i \left[ \cos(\alpha_i) A_i \cdot ((1 - \zeta_i) \cdot \boldsymbol{e} + 2\zeta_i \cos(\alpha_i) \cdot \boldsymbol{n}_i) \right]$$

$$= -\nu \frac{P_\odot}{m} \left( \frac{AU}{x} \right)^2 \cdot \sum_i \left[ \langle \boldsymbol{n}_i, \boldsymbol{e} \rangle A_i \cdot ((1 - \zeta_i) \cdot \boldsymbol{e} + 2\zeta_i \langle \boldsymbol{n}_i, \boldsymbol{e} \rangle \cdot \boldsymbol{n}_i) \right], \tag{A2}$$

where the sum is to be taken over all satellite surfaces $i$ that are illuminated by the sunlight, i.e. over all surfaces $i$ such that

$$\cos(\alpha_i) = \langle \boldsymbol{n}_i, \boldsymbol{e} \rangle > 0. \tag{A3}$$

Here, $\alpha_i$ is the angle of incidence, $\boldsymbol{n}_i$ is the outward pointing normal vector to the surface $i$, and $\boldsymbol{e}$ is the normalized vector pointing from the satellite's CoM towards the Sun. $x$ is the Sun-satellite distance, $A_i$ is the area of the surface $i$, so that $\cos(\alpha_i)A_i$ is its cross-sectional part. The $\zeta_i$ are the reflection coefficients of the respective surfaces, combining reflection co-

efficients for visible and IR light. $P_\odot$ denotes the solar radiation pressure at 1 AU (Astronomical Unit) distance from the Sun, with a flux (pressure times speed of light) amounting to about $1367\mathrm{Wm}^{-2}$. The left term under the sum in equation A2 accounts for the absorbed photons and the right term accounts for the photon reflections. The shadow function $\nu$ is a value between $0$ (in shadow) and $1$ (fully illuminated), calculated using a geometric shadow model with umbra and penumbra cones, ignoring atmosphere and flattening of the Earth. For more details look at Montenbruck and Gill (2000).

   In the above equations, the total mass for each GRACE-FO satellite is $m = 655$kg. A GRACE-FO satellite weighs about 180kg more than a GRACE satellite, due to the additional payload (Gath, 2016).

## Appendix B

There are several possible definitions of the pointing angles. However, if the rotation direction is clear, the different methods

differ only in the second order. I.e., the differences are in the order of microradians, at most, which can be considered negligible with respect to the measurement uncertainty.

   Inter-satellite pointing can be geometrically interpreted as deviations of the SF from the LOSF (Bandikova et al., 2012). Pointing jitter or variations can be expressed as a sequence of rotations about the roll (i.e, $\boldsymbol{x}_{LOSF}$), pitch (i.e, $\boldsymbol{y}_{LOSF}$) and yaw (i.e, $\boldsymbol{z}_{LOSF}$) axes. The roll, pitch and yaw angles can be derived from the matrix rotating from SF to LOSF (cf. Fig. 1).





The matrix rotating from SF to ICRF is related to the quaternions by (Wu et al., 2006):

$$\mathbf{R}_{SF}^{ICRF} = \begin{bmatrix} q_0^2 + q_1^2 - q_2^2 - q_3^2 & 2(q_1q_2 - q_0q_3) & 2(q_1q_3 + q_0q_2) \\ 2(q_1q_2 + q_0q_3) & q_0^2 - q_1^2 + q_2^2 - q_3^2 & 2(q_2q_3 - q_0q_1) \\ 2(q_1q_3 - q_0q_2) & 2(q_2q_3 + q_0q_1) & q_0^2 - q_1^2 - q_2^2 + q_3^2 \end{bmatrix} \tag{B1}$$

Here, $q$ are the quaternions mentioned in section 4.

The matrix rotating from ICRF to LOSF is derived from the orbital positions:

$$\mathbf{R}_{ICRF}^{LOSF} = [\boldsymbol{x}_{LOSF} \ \boldsymbol{y}_{LOSF} \ \boldsymbol{z}_{LOSF}], \tag{B2}$$

with the LOSF axes are column vectors according to the definition in section 2, expressed in inertial frame. Then, the pointing angles (roll $\theta_x$, pitch $\theta_y$ and yaw $\theta_z$) can be computed from the rotation matrix $\mathbf{R}_{SF}^{LOSF} = \mathbf{R}_{ICRF}^{LOSF} \cdot \mathbf{R}_{SF}^{ICRF}$ by:

$$\theta_x = \arctan\left(\frac{R_{32}}{R_{33}}\right)$$

$$\theta_y = -\arcsin\left(R_{31}\right)$$

$$\theta_z = \arctan\left(\frac{R_{21}}{R_{11}}\right) \tag{B3}$$

where $R_{ij}$ are the elements of $\mathbf{R}_{SF}^{LOSF}$. Here, the first index refers to the row and the second index refers to the column.

**Appendix C**

The following is a numerically stable pseudocode to compute quaternions from a given rotation matrix, where R denotes the rotation matrix and R(i,j) its element in the i[th] row and j[th] column.

```
IF ( R(1,1) >= R(2,2) AND R(1,1) >= R(3,3) )

    r = sqrt(1 + R(1,1) - R(2,2) - R(3,3));
    s = 1/(2*r);
    q0 = (R(3,2)-R(2,3))*s;
q1 = r/2;
    q2 = (R(2,1)+R(1,2))*s;
    q3 = (R(1,3)+R(3,1))*s;
```





```
   ELSEIF ( R(2,2) > R(1,1) AND R(2,2) >= R(3,3) )

   r = sqrt(1 + R(2,2) - R(1,1) - R(3,3));
   s = 1/(2*r);
q0 = (R(1,3)-R(3,1))*s;
   q1 = (R(1,2)+R(2,1))*s;
   q2 = r/2;
   q3 = (R(3,2)+R(2,3))*s;

ELSE

   r = sqrt(1 + R(3,3) - R(1,1) - R(2,2));
   s = 1/(2*r);
   q0 = (R(2,1)-R(1,2))*s;
q1 = (R(1,3)+R(3,1))*s;
   q2 = (R(2,3)+R(3,2))*s;
   q3 = r/2;

   END
```

**Appendix D**

Tables D1 and D2 describe the format of the data records for KBR1B and LRI1B simulated files. For the consistency we kept the tables similar to Case et al. (2002).



**Table D1.** KBR Data Format Record (KBR1B)

| Parameter | Definition | Format | Units |
|---|---|---|---|
| **gps_time** | GPS time, seconds past 12:00:00, noon 01-Jan-2000 | 9 i | s |
| **range** | Range between GRACE A and B | 16.10 f | m |
| **range_rate** | Range rate between GRACE A and B | 18.16 f | m/s |
| **range_accl** | Range acceleration between GRACE A and B | 21.18 f | $m/s^2$ |
| **ioni_corr** | Ionospheric range correction between GRACE A and B for Ka frequencies | 16.15 f | m |
| **lighttime_corr** | Light time range correction between GRACE A and B | 16.15 e | m |
| **lighttime_rate** | Light time range rate correction between GRACE A and B | 16.15 e | m/s |
| **lighttime_accl** | Light time range acceleration correction between GRACE A and B | 16.15 e | $m/s^2$ |
| **ant_centr_corr** | Antenna phase center range correction | 16.15 f | m |
| **ant_centr_rate** | Antenna phase center range rate correction | 16.15 e | m/s |
| **ant_centr_accl** | Antenna phase center range acceleration correction | 16.15 e | $m/s^2$ |
| **K_A_SNR** | SNR K band for GRACE A | 3 i | .1 db-Hz |
| **Ka_A_SNR** | SNR Ka band for GRACE A | 3 i | .1 db-Hz |
| **K_B_SNR** | SNR K band for GRACE B | 3 i | .1 db-Hz |
| **Ka_B_SNR** | SNR Ka band for GRACE B | 3 i | .1 db-Hz |
| **qualflg** | 0 = not Defined | 0.8 i | N/A |



**Table D2.** LRI Data Format Record (LRI1B)

| Parameter | Definition | Format | Units |
|---|---|---|---|
| **gps_time** | GPS time, seconds past 12:00:00, noon 01-Jan-2000 | 9 i | s |
| **range** | Range between GRACE A and B | 19.10 f | m |
| **range_rate** | Range rate between GRACE A and B | 19.16 f | m/s |
| **range_accl** | Range acceleration between GRACE A and B | 22.19 f | $m/s^2$ |
| **lighttime_corr** | Light time range correction between GRACE A and B | 16.15 e | m |
| **lighttime_rate** | Light time range rate correction between GRACE A and B | 16.15 e | m/s |
| **lighttime_accl** | Light time range acceleration correction between GRACE A and B | 16.15 e | $m/s^2$ |
| **ver_point_corr** | Vertex point range correction | 16.15 f | N/A |
| **ver_point_rate** | Vertex point range rate correction | 16.15 e | m/s |
| **ver_point_accl** | Vertex point range acceleration correction | 16.15 e | $m/s^2$ |
| **pitch_A_dws** | Pitch angle from differential wavefront sensing for GRACE A | 19.17 f | rad |
| **yaw_A_dws** | Yaw angle from differential wavefront sensing for GRACE A | 19.17 f | rad |
| **pitch_B_dws** | Pitch angle from differential wavefront sensing for GRACE B | 19.17 f | rad |
| **yaw_B_dws** | Yaw angle from differential wavefront sensing for GRACE B | 19.17 f | rad |
| **LRI_A_SNR** | SNR LRI for GRACE A | 3 i | .1 db-Hz |
| **LRI_B_SNR** | SNR LRI for GRACE B | 3 i | .1 db-Hz |
| **qualflg** | 0 = not Defined | 0.8 i | N/A |



*Author contributions.* M. Naeimi developed the orbit integrator code and performed the orbit simulations. H. Wegener and V. Müller developed the instrument noise models. G. Heinzel and M. Hewitson developed the LRI noise models. N. Darbeheshti performed the instrument noise simulations and prepared the manuscript with contributions from all co-authors.

*Competing interests.* The authors declare that they have no conflict of interest.

5   *Acknowledgements.* This project is supported by funding from the SFB 1128 "Relativistic Geodesy and Gravimetry with Quantum Sensors (geo-Q)" by the Deutsche Forschungsgemeinschaft. We also thank JPL and AIRBUS Defense and Space for providing the GRACE-FO attitude and orbital control system performance predictions.



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
