# Peer review of "Instrument Data Simulations for GRACE Follow-on: Observation and Noise Models"

_Earth System Science Data, 2017_

## Referee Comment (RC1) · Anonymous Referee #1 · 12 Sep 2017

General Comments

The GRACE mission has yielded much advancement in Earth system science thanks to the availability of a more than 15 years time series of mass transport data. Unfortunately, due to increasing problems with the battery on GRACE-2 the mission will shortly end and the hope is that GRACE Follow-on (GRACE-FO), due for launch early 2018, will be a worthy successor. As the name already says the prime objective of GRACE-FO will be to extend the GRACE time series using the same instruments and also data streams/formats. This would enable the users directly to start analyzing the new data.

The innovative addition on GRACE-FO will be a Laser Ranging Interferometer (LRI) which shall increase the accuracy of the inter-satellite measurement. In order to pre-

pare for GRACE-FO it is of high interest for a) gravity field processing centers to have a set of realistic instrument test data, especially for the new LRI, and b) for the user community to get a projection what the new LRI could provide for scientific applications when flown on GRACE-FO and what could be achieved on Next Generation Gravity Missions. Therefore the work done by the authors is of high interest especially for gravity field processing centers to prepare for using the GRACE-FO data.

The simulated instrument data set described in this paper is very significant, unique, useful, complete and worth to be published. Nethertheless, I provide some additional specific and technical comments as well as suggestions and some corrections which have to be taken into account in the next revision.

Specific Comments

- In the introduction section it is mentioned that the LRI data also provide information on attitude (pitch and yaw). I am not sure if this data is used/provided in the nominal GRACE-FO SDS Level-1B data processing. Some background information should be given.

- Same section: The authors mentioned Kim (2000) as a realistic pre-launch simulation of GRACE but forgot to mention Flechtner et al. (2016): What Can be Expected from the GRACE-FO Laser Ranging Interferometer for Earth Science Applications? - Surveys in Geophysics, 37, 2, p. 453-470. http://doi.org/10.1007/s10712-015-9338-y. Also here a simulation of the possible impact of the LRI on the GRACE-FO gravity field results and various applications is performed. In this paper 5 years of realistic instrument data and background model errors have been simulated (but not been made public).

- Also a hint to the plans of the GRACE-FO Science Data System to provide a set of simulated GRACE-FO data to the user community within the so called "Grand Simulation" shall be mentioned. E.g. taking the GRACE-FO status reports / abstracts of presentations in Kobe (IUGG) or Vienna (EGU). Nevertheless, it has to be mentioned that this activity is much behind schedule and therefore the provided data set is very

useful to start just now!

- It is not 100% clear what the time period of the simulated data is. I assume (without check by downloading the data) that it is one month. There are also later sentences such as "we simulated one month of data". This general information should be given at the very beginning.

- If it is just a month then analysis of trends or seasonal/sub-seasonal signals are not in the focus, but more a check if the simulated data can be read by the processing centers and how good the recovered field fits to the gravity field used for simulation. This points to question b) in the general comments. Should be discussed.

- Also I suggest that the authors provide some results at the end that they were successful with their own software (if existing?) to recover the noise-free and noisy data with such and such error (e.g. by degree variance plots). This would be close-loop verification before external users test the data.

- Page 3, Line 14: ...and a GPS error is added to each...: see comment below for KBR and SCA

- Page 3, Line 16: ...with added KBR errors...: I think here it should be already mentioned what is included in the error budget or at least a clear statement that the errors are all discussed in chapter 6.

- Page 4, Line 2: similar comment for the SCA1B errors

- Page 4, Line 3: For the Accelerometer data it looks like if they do not contain errors (but have as shown in Figure 2 and discussed later)

- Page 4, Line 7: same comment as for KBR1B

- Page 4, Figure 2: Imprecise, as quaternions here do not contain errors

- Page 5, Line 11: Don't understand "a static gravity field of d/o between 75 and 90". What has been used for simulation? Is this description a hint what is stated in line 18

namely that the user shall try to solve for the right degree and order which fits best to the simulated field? I would more expect that they used a fixed degree and order (e.g. 90) with coefficients which are unknown to the user and provide this max d/o to the user.

- Page 5, Line 10: It should be discussed that other gravitational forces such as atmosphere and ocean short term mass variations or an ocean pole tide model are not used (for simplicity) and this simulation data set focuses on impact of instrument data errors.

- Page 5, Line 12/13: Reference for eot11a and DE405 missing

- Page 6, Line 1ff: information on used altitude, eccentricity and inclination missing, also on length of the simulation (1 month?)

- Page 7, Line 12: Would be good to know (and also the reference) how large the GRACE noise level is for RPY angles.

- Page 10, Line 6: The authors mentioned ACC biases and scales. The value of the bias is known from Horwath et al (2011), but the values chosen for the scale factors are not known to the user. Some lines below it is only mentioned that certain constant values for each axis were chosen. As both have to be adjusted during gravity field determination it would be interesting to compare the simulated and adjusted values. The authors should mention if these scale factors will be made available.

- Page 10, Line 12ff: Here a first hint is given that one month of data was simulated (see comments above)

- Page 22, Line 2: This statement is not complete as the quality of GRACE-FO models also depends on knowledge of short-term tidal and non-tidal mass variations (which have also large influence on the adjusted gravity model; in contrast to Flechtner et al. was not simulated). Nevertheless, it is highly appreciated that such a well described set of simulated GRACE-FO instrument data is now being available to the different gravity

field processing centers.

Technical Corrections

- Page 6, Line 19: More importantly, even though the KBR will be the primary science instrument and the LRI, a technology demonstrator, a threshold: This sentence shall likely read: More importantly, even though the KBR will be the primary science instrument, for the LRI, a technology demonstrator, a threshold of...

- Page 11, Line 11: As the abbreviation is $\delta$SO I suggest to write ...is dominated by system and oscillator noise...

- Page 20, Line 9: Therefore, the angle determination is quantization limited shall read Therefore, the angle determination quantification is limited – or?

---

## Referee Comment (RC2) · Anonymous Referee #2 · 19 Sep 2017

Review of ESSD-2017-45, GRACE Follow-On

General comments: Very useful and timely data set. Well described in this paper. Very good product for publication in ESSD.

Specific comments:

Page 2 line 6: Simulator data for period of one month. But, if Grace FO continues monthly summary data as per GRACE, don't we need at least a two-month simulation to identify month-to-month variance of the monthly summaries?

Page 3 line 6: Why do we start with Figure 11 instead of Figure 1. This numbering remains a residual of originally having two separate manuscripts?

[Figure]

Page 3 line 9: Here we find actual resolution, daily at 5 seconds. Helpful to have this information earlier?

Page 4, line 8: Table 11 (and subsequently tables 21 and 31) instead of Table 1, 2, 3?

Comparing range, range rate and range acceleration noise predictions for KBR (Figure 17) and LRI (Figure 110) (AGAIN NOTE THE STRANGE NUMBERING SEQUENCE FOR FIGURES), (or likewise for time series in Figure 18 for KBR and Figure 112 for LRI) the authors suggest at least 2 order of magnitude lower noise, and in some cases perhaps better than 4 order of magnitude lower noise for the LRI. But this substantial improvement assumes, e.g. as described on page 5, that the laser ranging instrument pointing angle uncertainty - by engineering mechanism not yet solved - does not exceed some threshold which causes interferometry to fail ("falling out of lock"). The reader sees very hopeful numbers from this particular simulation but based on a very large assumption?

The two-orbit (roughly 3 hour) plots (Figure 14, Figure 19, Figure 114, 115, 117) provide the reader / user with highest resolution examples of specific per-orbit angles or jitter as reproduced by the simulations. But users applying Level-1B processing / formatting will not usually see or consider this level of detail?

---

## Author Comment (AC1) · 13 Oct 2017

Author's response to Referee 1

1. **Comments from the Anonymous Referee 1**
   - In the introduction section it is mentioned that the LRI data also provide information on attitude (pitch and yaw). I am not sure if this data is used/provided in the nominal GRACE-FO SDS Level-1B data processing. Some background information should be given.

   **Author's response**
   The following sentence was added in page 2: Therefore GRACE-FO LRI data processing will contain precise measurements of the satellites' pitch and yaw angles.

2. **Comments from the Anonymous Referee 1**
   Same section: The authors mentioned Kim (2000) as a realistic pre-launch simulation of GRACE but forgot to mention Flechtner et al. (2016): What Can be Expected from the GRACE-FO Laser Ranging Interferometer for Earth Science Applications? - Surveys in Geophysics, 37, 2, p. 453-470. http://doi.org/10.1007/s10712-015-9338-y. Also here a simulation of the possible impact of the LRI on the GRACE-FO gravity field results and various applications is performed. In this paper 5 years of realistic instrument data and background model errors have been simulated (but not been made public).

   **Author's response**
   The following sentence was added in page 1: Flechtner et al. (2016) have performed a full-scale simulation over the nominal GRACE-FO mission lifetime of 5 years and showed notable improvements with the LRI, on a global scale, of the order of 23%.

3. **Comments from the Anonymous Referee 1**
   Also a hint to the plans of the GRACE-FO Science Data System to provide a set of simulated GRACE-FO data to the user community within the so called ?Grand Simulation? shall be mentioned. E.g. taking the GRACE-FO status reports / abstracts of presentations in Kobe (IUGG) or Vienna (EGU). Nevertheless, it has to be mentioned that this activity is much behind schedule and therefore the provided data set is very useful to start just now!
   **Author's response**
   The following sentence was added in page 1: Also, GRACE-FO science data system team at Jet Propulsion Laboratory (JPL) has planned to release a GRACE-FO "Grand Simulation" data set before the real GRACE-FO data is available (Watkins et al., 2016).

4. **Comments from the Anonymous Referee 1**
   - It is not 100% clear what the time period of the simulated data is. I assume (without check by downloading the data) that it is one month. There are also later sentences such as ?we simulated one month of data?. This general information should be given at the very beginning.

   **Author's response**
   At page 2, line 6 in the **Introduction**, the duration of the simulated data has already been mentioned:
   'We have generated a set of simulated GRACE-FO data for the period of one month.'

5. **Comments from the Anonymous Referee 1**
   If it is just a month then analysis of trends or seasonal/sub-seasonal signals are not in the focus, but more a check if the simulated data can be read by the processing centers and how good the recovered field fits to the gravity field used for simulation. This points to question b) in the general comments. Should be discussed.
   **Author's response**
   The following paragraph was added in page 2: We have generated a set of simulated GRACE-FO data for a period of one month with 5-second sampling rate. A brief overview about the scope of the simulations are given in Naeimi et al. (2017).

The data set is available for download via https://doi.org/10.22027/AMDC2. The recovered gravity field solutions using this data set can be submitted via the same link. Therefore the goal of generating this set of simulated data are:

- Improving different gravity field recovery techniques, by comparing the input gravity field for the simulation and the recovered gravity fields
- Using new LRI data such as LRI ranging and LRI attitude information in different gravity field recovery techniques

The analysis of seasonal or sub-seasonal geophysical features are not the focus of this simulated data set, as the duration of the simulated data is short.

6. **Comments from the Anonymous Referee 1**
- Also I suggest that the authors provide some results at the end that they were successful with their own software (if existing?) to recover the noise-free and noisy data with such and such error (e.g. by degree variance plots). This would be close-loop verification before external users test the data.
**Author's response**
We did a close-loop verification. But as we mentioned in the introduction, the main scope of this paper is describing how the simulated data was generated. We will address the gravity field recovery from this set of simulated data by different approaches in a future paper.

7. **Comments from the Anonymous Referee 1**
Page 3, Line 14: . . .and a GPS error is added to each. . .: see comment below for KBR and SCA
**Author's response**
The following sentence was added in page 3: the following sections in this paper describe each simulated instrument observations and errors respectively:

8. **Comments from the Anonymous Referee 1**
- Page 3, Line 16: . . .with added KBR errors. . .: I think here it should be already mentioned what is included in the error budget or at least a clear statement that the errors are all discussed in chapter 6.
**Author's response**
The following sentence was added in page 3: the following sections in this paper describe each simulated instrument observations and errors respectively:

9. **Comments from the Anonymous Referee 1**
- Page 4, Line 2: similar comment for the SCA1B errors
**Author's response**
The following sentence was added in page 3: the following sections in this paper describe each simulated instrument observations and errors respectively:

10. **Comments from the Anonymous Referee 1**
- Page 4, Line 3: For the Accelerometer data it looks like if they do not contain errors (but have as shown in Figure 2 and discussed later)
**Author's response**
The following sentence was added in page 4: Then accelerometer noise, scale and bias are added.

11. **Comments from the Anonymous Referee 1**
- Page 4, Line 7: same comment as for KBR1B
**Author's response**
The following sentence was added in page 3: the following sections in this paper describe each simulated instrument observations and errors respectively:

12. **Comments from the Anonymous Referee 1**
- Page 4, Figure 2: Imprecise, as quaternions here do not contain errors

**Author's response**

The following sentence was added in Figure 2's label: please refer to Fig. 6 for detailed description on SCA simulated data

13. **Comments from the Anonymous Referee 1**

Page 5, Line 11: Don't understand ?a static gravity field of d/o between 75 and 90?. What has been used for simulation? Is this description a hint what is stated in line 18 namely that the user shall try to solve for the right degree and order which fits best to the simulated field? I would more expect that they used a fixed degree and order (e.g. 90) with coefficients which are unknown to the user and provide this max d/o to the user.

**Author's response**

The sentence was changed into: 'A static gravity field of a certain degree and order.' and the following sentence was added in page 6: 'The degree and order that was used as input are between 75 and 95.'

14. **Comments from the Anonymous Referee 1**

Page 5, Line 10: It should be discussed that other gravitational forces such as atmosphere and ocean short term mass variations or an ocean pole tide model are not used (for simplicity) and this simulation data set focuses on impact of instrument data errors.

**Author's response**

The following sentence was added in page 6: Other gravitational forces such as atmosphere and ocean short term mass variations are not used as this simulation data set focuses on impact of instrument data errors.

15. **Comments from the Anonymous Referee 1**

Page 5, Line 12/13: Reference for eot11a and DE405 missing

**Author's response**

The references, (Rieser et al., 2012) and (Standish, 1998) were added.

16. **Comments from the Anonymous Referee 1**

Page 6, Line 1ff: information on used altitude, eccentricity and inclination missing, also on length of the simulation (1 month?)

**Author's response**

This information were added in Table 1.

17. **Comments from the Anonymous Referee 1**

- Page 7, Line 12: Would be good to know (and also the reference) how large the GRACE noise level is for RPY angles.

**Author's response**

Th following sentence and Figure 4 were added in page 8: The GRACE star cameras are strong in the roll axis and weak in the pitch and yaw axes due to the orientation in which they were mounted (cf. Harvey, 2016). GRACE data (Fig.4) confirms $150 - 300 \frac{\mu\text{rad}}{\sqrt{\text{Hz}}}$ accuracy for pitch and yaw, and $25 - 35 \frac{\mu\text{rad}}{\sqrt{\text{Hz}}}$ for roll, which meets the mission requirements (cf. Stanton, 1998).

18. **Comments from the Anonymous Referee 1**

- Page 10, Line 6: The authors mentioned ACC biases and scales. The value of the bias is known from Horwath et al (2011), but the values chosen for the scale factors are not known to the user. Some lines below it is only mentioned that certain constant values for each axis were chosen. As both have to be adjusted during gravity field determination it would be interesting to compare the simulated and adjusted values. The authors should mention if these scale factors will be made available.

**Author's response**

The following sentence was added in page 11: The scale and bias parameters will be available via https://doi.org/10.22027/AMDC2 for comparison with the estimated ones.

19. **Comments from the Anonymous Referee 1**

- Page 10, Line 12ff: Here a first hint is given that one month of data was simulated (see comments above)

**Author's response**

This was addressed in item 4.

20. **Comments from the Anonymous Referee 1**

Page 22, Line 2: This statement is not complete as the quality of GRACE-FO models also depends on knowledge of short-term tidal and non-tidal mass variations (which have also large influence on the adjusted gravity model; in contrast to Flechtner et al. was not simulated). Nevertheless, it is highly appreciated that such a well described set of simulated GRACE-FO instrument data is now being available to the different gravity field processing centers.

**Author's response**

The sentence was changed into: We have described the simulation of observation and noise models for GRACE-FO multi-sensor system, consisting of inter-satellite ranging with microwave and laser ranging instrument, GPS orbit tracking, accelerometry, and attitude sensing.

21. **Comments from the Anonymous Referee 1**

Page 6, Line 19: More importantly, even though the KBR will be the primary science instrument and the LRI, a technology demonstrator, a threshold: This sentence shall likely read: More importantly, even though the KBR will be the primary science instrument, for the LRI, a technology demonstrator, a threshold of. . .

**Author's response**

The sentence was changed accordingly:

22. **Comments from the Anonymous Referee 1**

- Page 11, Line 11: As the abbreviation is $\delta_{SO}$ I suggest to write . . .is dominated by system and oscillator noise. . .

**Author's response**

The sentence was changed accordingly:

23. **Comments from the Anonymous Referee 1**

- Page 20, Line 9: Therefore, the angle determination is quantization limited shall read Therefore, the angle determination quantification is limited ? or?

**Author's response**

The sentence was changed into: However, the steering mirror can only turn in discrete units of $4.5\mu$rad around the pitch axis and $6\mu$rad around the yaw axis. Therefore, the angle determination is limited to integer multiples of these units.

Author's response to Referee 2

1. **Comments from the Anonymous Referee 2**

Page 2 line 6: Simulator data for period of one month. But, if Grace FO continues monthly summary data as per GRACE, don?t we need at least a two-month simulation to identify month-to-month variance of the monthly summaries?

**Author's response**

The following paragraph was added in page 2: Therefore the goal of generating this set of simulated data are:

– Improving different gravity field recovery techniques, by comparing the input gravity field for the simulation and the recovered gravity fields

– Using new LRI data such as LRI ranging and LRI attitude information in different gravity field recovery techniques

The analysis of seasonal or sub-seasonal geophysical features are not the focus of this simulated data set, as the duration of the simulated data is short.

2. **Comments from the Anonymous Referee 2**

Page 3 line 6: Why do we start with Figure 11 instead of Figure 1. This numbering remains a residual of originally

having two separate manuscripts?

**Author's response**

The Figure numbering is correct and starts from 1 in on-line PDF version.

3. **Comments from the Anonymous Referee 2**

Page 3 line 9: Here we find actual resolution, daily at 5 seconds. Helpful to have this information earlier?

**Author's response**

The following paragraph was added in page 2: We have generated a set of simulated GRACE-FO data for a period of one month with 5-second sampling rate.

4. **Comments from the Anonymous Referee 2**

Page 4, line 8: Table 11 (and subsequently tables 21 and 31) instead of Table 1, 2, 3?

**Author's response**

The Table numbering is correct and starts from 1 in on-line PDF version.

5. **Comments from the Anonymous Referee 2**

Comparing range, range rate and range acceleration noise predictions for KBR (Figure 17) and LRI (Figure 110) (AGAIN NOTE THE STRANGE NUMBERING SEQUENCE FOR FIGURES), (or likewise for time series in Figure 18 for KBR and Figure 112 for LRI) the authors suggest at least 2 order of magnitude lower noise, and in some cases perhaps better than 4 order of magnitude lower noise for the LRI. But this substantial improvement assumes, e.g. as described on page 5, that the laser ranging instrument pointing angle uncertainty - by engineering mechanism not yet solved - does not exceed some threshold which causes interferometry to fail (?falling out of lock?). The reader sees very hopeful numbers from this particular simulation but based on a very large assumption?

**Author's response**

We think that the paragraph in page 6 lines 17 to 23 is misleading, so we reformulated it into:

An attitude and orbit control system keeps the satellite orientation near its nominal attitude, within a certain boundary for each of the three pointing angles. These boundaries have been lowered for GRACE-FO compared to GRACE for two reasons. Firstly, due to the coupling of pointing angle errors into the ranging data; experience has shown that improved pointing would enhance the quality of gravity field solutions (Horwath et al., 2011). Secondly, the LRI requires better satellite pointing, in order to guarantee its functionality; otherwise there is a risk that the laser beam starts to hit obstacles. Hence, the combined effect of pointing jitter on one hand and frame misalignments on the other hand, together, cannot exceed a certain value (of about a few milliradians in terms of pitch and yaw angles for GRACE-FO). This yields strict requirements for the construction and mounting of the LRI components, and also the necessity for an improved pointing control.

The pointing jitter angles describe how the 'true' satellite orientation (as it actually is) deviates from the 'nominal' orientation (as it should be ideally in the absence of pointing angles). The nominal orientation is satellites' attitude reference. We assumed the satellites' attitude reference is the alignment of SF and LOSF for the simulations.

6. **Comments from the Anonymous Referee 2**

The two-orbit (roughly 3 hour) plots (Figure 14, Figure 19, Figure 114, 115, 117) provide the reader / user with highest resolution examples of specific per-orbit angles or jitter as reproduced by the simulations. But users applying Level-1B processing / formatting will not usually see or consider this level of detail?

**Author's response**

Yes, but in this section we are referring to two references, Bandikova et al. (2012) and Horwath et al. (2011). In both, there are similar attitude information plots for three hours orbit. Accordingly, the following paragraph was added in page 8:

The value of bias for each angle was chosen in range of a few milliradians. This level of bias has investigated by Horwath

et al. (2011) based on GRACE Level-1B data. Fig. 5 shows simulated star camera roll, pitch and yaw angles, which are similar to the GRACE inter-satellite pointing variations plot in Bandikova et al. (2012).

**Instrument Data Simulations for GRACE Follow-on: Observation and Noise Models**

Neda Darbeheshti[1], Henry Wegener[1], Vitali Müller[1], Majid Naeimi[2], Gerhard Heinzel[1], and
Martin Hewitson[1]

[1]Max Planck Institute for Gravitational Physics (Albert Einstein Institute)-Leibniz Universität Hannover
[2]Institut für Erdmessung-Leibniz Universität Hannover

*Correspondence to:* Neda Darbeheshti (neda.darbeheshti@aei.mpg.de)

**Abstract.** The Gravity Recovery and Climate Experiment (GRACE) mission has yielded data on the Earth's gravity field to monitor temporal changes for more than fifteen years now. The GRACE twin satellites use microwave ranging with micrometer precision to measure distance variations between two satellites caused by the Earth's global gravitational field. GRACE Follow-on (GRACE-FO) will be the first satellite mission to use inter-satellite laser interferometry in space. The laser ranging instrument (LRI) will provide two additional measurements compared to the GRACE mission: interferometric inter-satellite ranging with nanometer precision and inter-satellite pointing information. We have designed a set of simulated GRACE-FO data, which include LRI measurements, apart from all other GRACE instrument data needed for the Earth's gravity field recovery. The simulated data files are publicly available via https://doi.org/10.22027/AMDC2 and can be used to derive gravity field solutions like from GRACE data. This paper describes the scientific basis and technical approaches used to simulate the GRACE-FO instrument data.

*Copyright statement.* TEXT

**1 Introduction**

The space gravimetry mission GRACE (Tapley et al., 2004) observes the Earth's gravity field changes with time. GRACE is the first low-low satellite-to-satellite tracking mission: the principal measurement is the distance variability between low orbit GRACE twin satellites which translates into the monthly gravity models (Wahr et al., 1998).

Kim (2000) published the first GRACE satellite simulation study before the launch of the GRACE satellites (in 2002). Now, seventeen years later, GRACE satellites are at the end of their lifetime and GRACE-FO data will be available soon. Although the GRACE-FO mission, and respectively its instrument data streams, will be very similar to GRACE, the necessity for GRACE-FO instrument data simulation emerges from the additional interferometric inter-satellite ranging. Flechtner et al. (2016) have performed a full-scale simulation over the nominal GRACE-FO mission lifetime of 5 years and showed notable improvements with the LRI, on a global scale, of the order of 23%. Also, GRACE-FO science data system team at Jet Propulsion Laboratory

(JPL) has planned to release a GRACE-FO "Grand Simulation" data set before the real GRACE-FO data is available (Watkins et al., 2016).

Most importantly, the operation of the LRI in addition to the primary K-band ranging (KBR) instrument yields extra information not only in the ranging measurement, but also in the attitude determination, since the LRI data stream. Therefore

5  GRACE-FO LRI data processing will contain precise measurements of the satellites' pitch and yaw angles. In this paper for the first time, simulated LRI pitch and yaw angles are provided. Exploitation of the new GRACE-FO measurements has great potential to improve spatial and temporal resolution of the Earth's gravity field solutions.

Also, there are different techniques to recover the Earth's gravity field from GRACE-like data (e.g., Reigber (1989), Gerlach et al. (2003), Mayer-Gürr (2006) , Rummel (1979)). Therefore, simulated instrument data provide a controlled, closed form

10  medium, to test and improve different gravity field recovery techniques.

We have generated a set of simulated GRACE-FO data for the a period of one month with 5-second sampling rate. A brief overview about the scope of the simulations are given in Naeimi et al. (2017). The data set is available for download via https://doi.org/10.22027/AMDC2. The recovered gravity field solutions using this data set can be submitted via the same link. Therefore the goal of generating this set of simulated data are:

15  – Improving different gravity field recovery techniques, by comparing the input gravity field for the simulation and the recovered gravity fields

– Using new LRI data such as LRI ranging and LRI attitude information in different gravity field recovery techniques

The analysis of seasonal or sub-seasonal geophysical features are not the focus of this simulated data set, as the duration of the simulated data is short.

20  The main purpose of this paper is to describe the chain of instrument data simulation procedure. The first section presents the preliminaries for the data simulation, including the coordinates systems and symbols, followed by each section describing each instrument data simulation, including details of instruments' noise models.

**2  Preliminaries**

The following coordinate systems are used to define the various simulated data:

25  International Celestial Reference Frame (ICRF) – Inertial frame:

– origin: center of mass (CoM) of the Earth

– axes: according to IERS 2010 conventions (Petit and Luzum, 2010)

International Terrestrial Reference Frame (ITRF) – Earth-fixed (co-rotating) frame:

30  – origin: CoM of the Earth

– axes: according to IERS 2010 conventions (Petit and Luzum, 2010)

line-of-sight frame (LOSF), one per satellite, for GRACE A:

– origin: satellite's CoM

5    – $x_{LOSF_A} = \frac{r_B - r_A}{|r_B - r_A|}$, where $r$ is the satellites' position vector in the ICRF (i.e., line-of-sight vector and roll axis).

– $y_{LOSF_A} = \frac{x_{LOSF_A} \times r_A}{|x_{LOSF_A} \times r_A|}$ (i.e., pitch axis)

– $z_{LOSF_A} = x_{LOSF_A} \times y_{LOSF_A}$ (i.e., yaw axis)

(for GRACE B, A and B indices should be exchanged.)

10   satellite frame (SF), one per satellite according to Case et al. (2002):

– origin: satellite's CoM

– $x_{SF}$ = from the origin to a target location of the phase center of the K/Ka band horn

– $y_{SF}$ = forms a right-handed triad with $x_{SF}$ and $z_{SF}$

– $z_{SF}$ = normal to $x_{SF}$ and to the plane of the main equipment platform, and positive towards the satellite radiator

15     on the bottom of the GRACE-FO

The LOSF and SF are shown in Fig. 1. Since we did not model variations of the satellites' CoM (and the CoM coinciding with the on-board accelerometer's proof masses) for data simulation, the SF coincides with the science reference frame defined in Case et al. (2002).

20   All simulated data are published in GRACE Level-1B data format: daily files with 5-second sampling rate (cf. Case et al., 2002). They can be considered pre-processed like GRACE Level-1B data. Time tags are given in GRACE GPS seconds, i.e. seconds since epoch 2000-01-01, 12:00:00 (no leap seconds applied). Five instrument data types were simulated; the following sections in this paper describe each simulated instrument observations and errors respectively:

– GPS Navigation Data (GNV1B)

25    Simulated GPS positions and velocities are the output of the orbit integrator, which are rotated from ICRF to ITRF, and a GPS error is added to each. The error-free positions can be considered a kinematic orbit.

– K-Band Ranging System (KBR1B)

Simulated KBR ranging data is derived from the error-free GPS positions and velocities with added KBR errors.

– Star Camera (SCA1B)

30    Simulated star camera quaternions are derived from the simulated roll, pitch and yaw angles with added errors.

[Figure]

[Figure]

**Figure 1.** Illustration of SF and LOSF for GRACE satellites. Small positive yaw (left) and pitch (right) angles indicate the direction of rotation defining the sign of the pointing angles

– Accelerometer (ACC1B)

Simulated linear accelerations are calculated from the non-gravitational accelerations acting on the satellites. The error-free simulated star camera quaternions are used to transform the linear accelerations from ICRF to SF. Then accelerometer noise, scale and bias are added. The angular accelerations are calculated from the error-free simulated star camera quaternions.

– Laser Ranging Instrument (LRI1B)

Simulated LRI ranging data is derived from error-free GPS positions and velocities with added LRI errors.

Figure Fig. 2 shows a flowchart of the procedure used for the simulations. For each instrument, first the error-free observation was generated, and then the errors including instrument noise, bias and scale were applied to each instrument observation.

In this paper,

– The symbols $\delta$ and $\Delta$ are used for time-varying and constant errors, respectively.

– The symbol $\tilde{\delta}$ denotes amplitude spectral densities (ASD).

For data simulations,

– A five points numerical differentiation method was used for the numerical differentiations.

[Figure]

**Figure 2.** Flowchart of the simulation steps for GRACE-FO instrument data; please refer to Fig. 6 for detailed description on SCA simulated data

- The LISA Technology Package Data Analysis (LTPDA) toolbox (https://www.elisascience.org/ltpda/) for MATLAB was used for generation of time series based on instrument noise models given in terms of ASD. LTPDA uses Franklin's random noise generator method (Franklin, 1965) to generate arbitrarily long time series with a prescribed spectral density.

**3 Simulating GNV1B Data**

An orbit integrator is used to calculate the trajectories of the GRACE-FO satellites (GRACE-FO A and GRACE-FO B) by numerical integration of Newton's second law of motion, based on knowledge of the forces acting on the satellite. Table 1 summarises the orbit integrator parameters.

The IERS2010 conventions are used for rotation between the ITRF and the international celestial reference frame ICRF. Two types of force models were used for orbit integration:

Gravitational forces:

- A static gravity field of a certain degree and order.
- The Ocean tide model  EOT11a (Rieser et al., 2012) up to degree and order 80.
- Direct tides of the Moon and Sun using NASA  JPL DE405 ephemeris (Standish, 1998).
- Anelastic solid Earth tides according to IERS2010.

Non-gravitational forces:

**Table 1.** Orbit integrator parameters

| Parameter | Description |
|---|---|
| Altitude | 477.7 km |
| Eccentricity |  0.0019 |
| Inclination | $89.0081°$ |
|  Numerical integration approach |  Gauss-Jackson order 12 |
| Integration length | 31 days (May 2005) |
| Integration step size | 5 seconds |

- Atmospheric drag model

- Solar radiation pressure model

The static gravity model and its exact degree and order are the unknowns for the gravity field recovery. The degree and order that was used as input are between 75 and 95. The atmospheric drag and solar radiation pressure models are described in Appendix

5   A. Other gravitational forces such as atmosphere and ocean short term mass variations are not used as this simulation data set focuses on impact of instrument data errors.

The input to the orbit integrator is the initial time and state (position and velocity vectors) of GRACE-FO A and GRACE-FO B at time 00:00:00, 2005-05-01. It calculates the two trajectories separately, beside the time series of accelerations along the trajectory from the gravitational and non gravitational force models. The output of the orbit integrator are the time series of the

10   position, velocity and acceleration vectors of GRACE-FO A and GRACE-FO B:

$$\boldsymbol{r}_A, \quad \dot{\boldsymbol{r}}_A, \quad \ddot{\boldsymbol{r}}_A, \quad \boldsymbol{r}_B, \quad \dot{\boldsymbol{r}}_B, \quad \ddot{\boldsymbol{r}}_B$$

White noise with a level of a few $\frac{\text{cm}}{\sqrt{\text{Hz}}}$ was generated along $x$, $y$ and $z$ axes independently, and added to each satellite position:

$$\boldsymbol{r}_{GNV1B} = \boldsymbol{r} + \delta\boldsymbol{r}_{GNV1B} \tag{1}$$

15   Then the noise was differentiated numerically and added to the velocities along $x$, $y$ and $z$ axes separately for each satellite:

$$\dot{\boldsymbol{r}}_{GNV1B} = \dot{\boldsymbol{r}} + \delta\dot{\boldsymbol{r}}_{GNV1B} \tag{2}$$

**4   Simulating SCA1B Data**

The satellite attitude with respect to the ICRF is determined from the star cameras on board the satellites. The measured attitude is expressed in terms of quaternions $q$:

20   $$q = \begin{pmatrix} q_0 & q_1 & q_2 & q_3 \end{pmatrix} \tag{3}$$

Here, $q_0$ denotes the real component and $q_1$, $q_2$ and $q_3$ are the imaginary components of the quaternion. The time series of quaternions is provided in the SCA1B product.

 An attitude and orbit control system keeps the satellite orientation near its nominal attitude, within a certain boundary for each of the three pointing angles. These boundaries have been lowered for GRACE-FO compared to GRACE for two reasons. Firstly, due to the  coupling of pointing angle errors into the ranging data; experience has shown that improved pointing would enhance the quality of gravity field solutions (Horwath et al., 2011). Secondly, the LRI  requires better satellite pointing, in order to guarantee its functionality; otherwise there is a risk that the laser beam starts to hit obstacles. Hence, the  combined effect of pointing jitter on one hand and frame misalignments on the other hand, together, cannot exceed a certain value (of about a few milliradians in terms of pitch and yaw angles for GRACE-FO). This yields strict requirements for the construction and mounting of the LRI components, and also the necessity for an improved pointing control.

The pointing jitter angles describe how the 'true' satellite orientation (as it actually is) deviates from the 'nominal' orientation (as it should be ideally in the absence of pointing angles). The nominal orientation is satellites' attitude reference. We assumed the satellites' attitude reference is the alignment of SF and LOSF for the simulations.

[Figure]

**Figure 3.** ASD of simulated roll, pitch and yaw angles

Accordingly, satellite pointing angles can be computed from star camera quaternions and orbital positions (described in Appendix B). For simulating star camera quaternions, one has to go the opposite way. Pointing angles from GRACE-FO attitude and orbital control system performance predictions were provided to us by JPL and AIRBUS Defense and Space. A model, which is based on the spectrum of these predicted angles, was used to simulate the pointing angles. The common approach for generating time series with a known spectrum is to use a random noise generator.  Fig. 3 shows the ASD of the simulated roll ($\theta_x$), pitch ($\theta_y$) and yaw ($\theta_z$) angles. One can see that all three angles have peaks mostly in the frequency band between $10^{-4}$ and $2 \cdot 10^{-3}$. These peaks disturb the functionality of the random noise generator, thus they were modelled individually. The result is a time series of error-free inter-satellite pointing angles.

To simulate star camera measurements, white noise ($\delta\theta_{SCA1B}$) and a bias ($\Delta\theta_{SCA1B}$) was added to each error-free angle separately:

$$\theta_{x,SCA1B} = \theta_x + \delta\theta_{x,SCA1B} + \Delta\theta_{x,SCA1B}$$

$$\theta_{y,SCA1B} = \theta_y + \delta\theta_{y,SCA1B} + \Delta\theta_{y,SCA1B}$$

$$\theta_{z,SCA1B} = \theta_z + \delta\theta_{z,SCA1B} + \Delta\theta_{z,SCA1B} \tag{4}$$

Here, $\theta_x$, $\theta_y$ and $\theta_z$ are the error-free simulated roll, pitch and yaw angles; $\theta_{x,SCA}$, $\theta_{y,SCA}$ and $\theta_{z,SCA}$ are simulated star camera roll, pitch and yaw angles.

The GRACE-FO satellites are equipped with improved star cameras compared to GRACE, and the number of star camera heads will increase from two to three per satellite (Gath, 2016); also Bandikova et al. (2012) suggested that proper combination of the different star camera heads reduces high frequency noise of the pointing angles. Accordingly, it is expected that a better estimation of pointing angles from GRACE-FO star camera data will be available. Therefore, white noise with a level of a few ten $\frac{\mu rad}{\sqrt{Hz}}$ was chosen, which is lower than the current noise level in roll, pitch and yaw  angles estimated from the GRACE star camera data. The GRACE star cameras are strong in the roll axis and weak in the pitch and yaw axes due to the orientation in which they were mounted (cf. Harvey, 2016). GRACE data (Fig. 4) confirms $150 - 300 \frac{\mu rad}{\sqrt{Hz}}$ accuracy for pitch and yaw, and $25 - 35 \frac{\mu rad}{\sqrt{Hz}}$ for roll, which meets the mission requirements (cf. Stanton et al., 1998).

The value of bias for each angle was chosen in range of a few milliradians . This level of bias has investigated by Horwath et al. (2011) based on GRACE Level-1B data. Fig. 5 shows simulated star camera roll, pitch and yaw angles, which are similar to the GRACE inter-satellite pointing variations plot in Bandikova et al. (2012).

From the contaminated simulated pointing angles of equations (4), the rotation matrix $\mathbf{R}_{SF}^{LOSF}$ was built; and with the error-free simulated orbit positions, the rotation matrix $\mathbf{R}_{LOSF}^{ICRF}$ was built; having these two matrices, the matrix $\mathbf{R}_{SF}^{ICRF}$ is

$$\mathbf{R}_{SF}^{ICRF} = \mathbf{R}_{LOSF}^{ICRF} \cdot \mathbf{R}_{SF}^{LOSF}, \tag{5}$$

[Figure]

**Figure 4.** ASD of GRACE roll, pitch and yaw angles on 2008-12-01

containing the simulated star camera quaternions (cf. Fig. 6). Finally, the simulated quaternions can be recovered from the rotation matrix $\mathbf{R}_{SF}^{ICRF}$ by using the equations (Wu et al., 2006):

$$
\begin{aligned}
q_0 &= \frac{1}{2}\sqrt{1 + R_{11} + R_{22} + R_{33}} \\
q_1 &= \frac{(R_{23} - R_{32})}{4q_0} \\
q_2 &= \frac{(R_{31} - R_{13})}{4q_0} \\
q_3 &= \frac{(R_{12} - R_{21})}{4q_0}
\end{aligned}
\tag{6}
$$

where $R_{ij}$ are the elements of $\mathbf{R}_{SF}^{ICRF}$. Note that the equations 6 are only numerically stable, as long as the trace of $\mathbf{R}$ is non-negative (i.e. not close to $-1$). A numerically stable pseudocode that was used is shown in Appendix C.

Two other sets of quaternions were generated. Error-free quaternions from error-free pointing angles in equations (4); and noisy quaternions that come from white noise contaminated pointing angles without the bias. We will refer to these two set of quaternions in the following sections.

**5 Simulating ACC1B Data**

 Fig. 2 shows that the non-gravitational accelerations were computed along the orbit in ICRF.

[Figure]

**Figure 5.** Simulated star camera roll, pitch and yaw angles during two orbital revolutions for GRACE-FO A

**5.1 Linear Accelerations**

The non-gravitational accelerations are the sum of atmospheric drag and solar radiation pressure accelerations (cf. Appendix A) along the orbit in inertial frame (ICRF). The non-gravitational accelerations $\ddot{r}^{ICRF}$ were transformed into the satellite frame $\ddot{r}^{SF}$ using the rotation matrix $\mathbf{R}^{SF}_{ICRF}$ from error-free simulated quaternions:

$$\ddot{r}^{SF} = \mathbf{R}^{SF}_{ICRF} \cdot \ddot{r}^{ICRF} \tag{7}$$

After being transformed into the SF, the linear accelerations were multiplied by the scale factors $s_x$, $s_x$ and $s_z$, and then the accelerometer noise time series ($\delta\ddot{r}_{ACC1B}$) and the biases ($\Delta\ddot{r}_{ACC1B}$) were added along $x$, $y$ and $z$ axes independently:

$$\ddot{r}^{SF}_{ACC1B} = \begin{bmatrix} s_x & 0 & 0 \\ 0 & s_y & 0 \\ 0 & 0 & s_z \end{bmatrix} \cdot \ddot{r}^{SF} + \delta\ddot{r}_{ACC1B} + \Delta\ddot{r}_{ACC1B} \tag{8}$$

[Figure]

**Figure 6.** Flowchart of the simulation steps for SCA1B data

The ASD noise model of Kim (2000) was used to generate accelerometer noise ($\delta\ddot{\boldsymbol{r}}_{ACC1B}$):

$$\tilde{\delta\ddot{\boldsymbol{r}}}_{x/z,ACC1B}(f) = 10^{-10} \cdot \sqrt{1 + \frac{0.005\text{Hz}}{f}} \, \frac{\text{m/s}^2}{\sqrt{\text{Hz}}} \qquad 10^{-5} \leq f \leq 10^{-1}$$

$$\tilde{\delta\ddot{\boldsymbol{r}}}_{y,ACC1B}(f) = 10^{-9} \cdot \sqrt{1 + \frac{0.1\text{Hz}}{f}} \, \frac{\text{m/s}^2}{\sqrt{\text{Hz}}} \qquad 10^{-5} \leq f \leq 10^{-1}$$

(9)

The $y$ axis in SF ($\boldsymbol{y}_{SF}$ in Fig. 1) is considered the least sensitive axis for accelerometer measurements (Kim, 2000). The noise ASD of the sensitive axes and the less sensitive axis are shown in Fig. 7. One month time series of the accelerometer noise

5    was generated separately for $x$, $y$ and $z$ axes from the ASD models and added to the accelerations (equation (8)). Values close to GRACE accelerometer scale and bias along each axis were chosen, and kept constant for the one month of the simulated data. Therefore, in total for both satellites, six accelerometer scale parameters and six accelerometer bias parameters should be estimated during the gravity field recovery using one month of the simulated data. The scale and bias parameters will be available via https://doi.org/10.22027/AMDC2 for comparison with the estimated ones.

[Figure]

**Figure 7.** ASD of accelerometer noise

[revised manuscript text omitted]

5   Flechtner, F., Neumayer, K.-H., Dahle, C., Dobslaw, H., Fagiolini, E., Raimondo, J.-C., and Güntner, A.: What Can be Expected from the GRACE-FO Laser Ranging Interferometer for Earth Science Applications?, Surveys in Geophysics, 37, 453–470, doi:10.1007/s10712-015-9338-y, https://doi.org/10.1007/s10712-015-9338-y, 2016.

Franklin, J. N.: Numerical simulation of stationary and non-stationary gaussian random processes, SIAM Review, 7, 68–80, 1965.

Gath, P.: Integration und Test der GRACE Follow-On Satelliten, Tech. Rep. 420305, Deutscher Luft- und Raumfahrtkongress, 2016.

10  Gerlach, C., Földvary, L., Švehla, D., Gruber, T., Wermuth, M., Sneeuw, N., Frommknecht, B., Oberndorfer, H., Peters, T., Rothacher, M., Rummel, R., and Steigenberger, P.: A CHAMP-only gravity field model from kinematic orbits using the energy integral, Geophysical Research Letters, 30, 2003.

Harvey, N.: GRACE star camera noise, Advances in Space Research, 58, 2016.

Horwath, M., Lemoine, J., Biancale, R., and Bourgogne, S.: Improved GRACE science results after adjustment of geometric biases in the Level-1B K-band ranging data, Journal of Geodesy, 85, 2011.

15  Kim, J.: Simulation study of a low-low satellite-to-satellite tracking mission, Ph.D. thesis, The University of Texas at Austin, 2000.

Mayer-Gürr, T.: Gravitationsfeldbestimmung aus der Analyse kurzer Bahnbögen am Beispiel der Satellitenmissionen CHAMP und GRACE, Ph.D. thesis, Universitäts-und Landesbibliothek Bonn, 2006.

Montenbruck, O. and Gill, E.: Satellite Orbits, "Springer-Verlag, "Berlin Heidelberg 2000, 2000.

20  Müller, V.: Orbit simulation Toolkit, Ph.D. thesis, Bachelor of Science Degree Thesis, Max-Planck-Institute for Gravitational Physics (Albert Einstein Institute, AEI) and Gottfried Wilhelm Leibniz Universtitut, Hannover, 2010.

Naeimi, M., Wegener, H., Darbeheshti, N., Mueller, V., Schnitger, A., Goswami, S., Flury, J., Hewitson, M., and Heinzel, G.: Instrument Data Simulations for GRACE Follow-on: Overview and Goals, Advances in Space Research, under review, 2017.

Petit, G. and Luzum, B.: IERS Conventions (2010), Tech. Rep. IERS Technical Note No. 36, Frankfurt am Main: Verlag des Bundesamts für Kartographie und Geodäsie, 2010.

25  Reigber, C.: Gravity field recovery from satellite tracking data, in: Theory of satellite geodesy and gravity field determination, pp. 197–234, Springer, 1989.

Rieser, D., Mayer-Gürr, T., Savcenko, R., Bosch, W., Wünsch, J., Dahle, C., and Flechtner, F.: The ocean tide model EOT11a in spherical harmonics representation, Technical Note, 2012.

30  Rummel, R.: Determination of short-wavelength components of the gravity field from satellite-to-satellite tracking or satellite gradiometry-an attempt to an identification of problem areas, Manuscripta Geodaetica, 4, 107–148, 1979.

Sheard, B., Heinzel, G., Danzmann, K., Shaddock, A., Klipstein, W., and Folkner, W.: Intersatellite laser ranging instrument for the GRACE follow-on mission, J Geod, 86, 1083–1095, 2012.

Standish, E. M.: JPL Planetary and Lunar Ephemerides, DE405/LE405, Jet Propulsion Laboratory Interoffice Memorandum, IOM 312.F - 35   98 - 048, 1998.

Stanton, R., Bettadpur, S., Dunn, C., Renner, K.-P., and Watkins, M.: Gravity Recovery And Climate Experiment (GRACE) science & mission requirements document, 1998.

Tapley, B. D., Bettadpur, S., Watkins, M., and Reigber, C.: The gravity recovery and climate experiment: Mission overview and early results, Geophysical Research Letters, 31, n/a–n/a, doi:10.1029/2004GL019920, http://dx.doi.org/10.1029/2004GL019920, l09607, 2004.

Wahr, J., Molenaar, M., and Bryan, F.: Time variability of the Earth's gravity field: Hydrological and oceanic effects and their possible detection using GRACE, Journal of Geophysical Research: Solid Earth, 103, 30 205–30 229, doi:10.1029/98JB02844, http://dx.doi.org/10.1029/98JB02844, 1998.

Watkins, M., Flechtner, F., Webb, F., Landerer, F., and Grunwald, L.: Current Status of the GRACE Follow-On Mission, Presented at the European Geosciences Union General Assembly, Vienna , Austria, 2016.

Wu, S.-C., Kruizinga, G., and Bertiger, W.: Algorithm theoretical basis document for GRACE level-1B data processing V1. 2, 2006.